# A Climate Data Record of Year-Round Global Sea Ice Drift from the EUMETSAT OSI SAF

Thomas Lavergne[1,*] and Emily Down[1,*]

[1]Research and Development Department, Norwegian Meteorological Institute, Oslo, Norway
[*]These authors contributed equally to this work.

**Correspondence:** Thomas Lavergne (thomas.lavergne@met.no)

**Abstract.**

Sea ice in the polar regions can move several tens of kilometers per day under the actions of winds, ocean currents, and internal stresses. Long-term observations of the rate and patterns of this motion are needed to characterize the full response of the polar environment to climate change. Here, we introduce a new climate data record (CDR) of year-round, global, daily sea-ice drift vectors covering 1991-2020. The motion vectors are computed from series of passive microwave imagery in the winter months, and from a parametric free-drift model in the summer months. An evaluation against on-ice buoy trajectories reveals that the RMSEs of the sea-ice drift CDR are small and vary with hemisphere and seasons (2.1 km for Arctic winters, 2.6 km for Arctic summer, 3 to 4 km for the Antarctic sea ice). The CDR is un-biased for Arctic winter conditions. The bias is larger for Antarctic and for summer sea ice motion. The CDR consists of daily product files holding the $\mathrm{d}X$ and $\mathrm{d}Y$ components of the drift vectors on an EASE2 grid with 75 km spacing, as well as associated uncertainties and flags. It is prepared in the context of the EUMETSAT Ocean and Sea Ice Satellite Application Facility (OSI SAF) and is readily available as https://doi.org/10.15770/EUM_SAF_OSI_0012 (OSI SAF, 2022).

## 1 Introduction

Sea ice in the polar regions can move several tens of kilometers per day under the actions of winds, ocean currents, and internal stresses. This motion redistributes sea ice from regions where it forms during winter, to regions where it melts during summer. Such redistribution of mass and fresh water has a profound influence on the polar oceans and their currents. Patterns of sea-ice motion define key characteristics of the sea-ice cover, such as the accumulation and re-circulation of thicker and older ice north for Canada and in the Beaufort Sea, or ice formation in latent heat polynyas. The distribution, export, and replenishment of sea-ice age is fully controlled by where sea ice drifts. At a smaller scale, sea-ice motion steers the thickness distribution of sea-ice through the opening of leads and the formation of ridges. Trends in sea-ice drift have been reported by several investigators, associated to area-reduction and thinning of the sea-ice cover in the Arctic (Rampal et al., 2009; Spreen et al., 2011; Kwok et al., 2013; Olason and Notz, 2014) and stronger winds over Antarctic sea-ice (Holland and Kwok, 2012; Kwok et al., 2017). These trends have consequences beyond sea ice itself, e.g. on the salinity of the Southern Ocean (Haumann et al.,

2016), the long-range sediment transport across the Arctic Ocean (Krumpen et al., 2019), or the movements and energy balance of polar bears (Durner et al., 2017) to name a few.

Owing to its such a fundamental role, error-characterized, multi-decadal observations of global sea-ice drift are required by the Implementation Plans of the Global Climate Observing System (GCOS), a body of the World Meteorological Organization (WMO) that assesses the maturity of the observing system required to monitor our changing climate and gives guidance for its development (GCOS IP-22; GCOS ECVReqs-22). Sea-ice drift is one of the seven variables defining the sea ice Essential Climate Variable (ECV) (Lavergne and Kern et al., 2022).

In the Arctic, trajectories from on-ice buoys and drifting platforms or ships have contributed at an early stage to our knowledge of the mean patterns of sea-ice motion (Nansen, 1897; Colony and Thorndike, 1984; Rigor et al., 2002) and its trends (Rampal et al., 2009; Olason and Notz, 2014). This is mostly thanks to the systematic deployment and data stewardship of on-ice buoys by the International Arctic Buoy Programme (IABP), whose first array of buoys were deployed in the late 1970s (Thorndike, 1980). Similar buoy coordination efforts were later started for Antarctic sea ice through the International Programme for Antarctic Buoys (IPAB), but the fewer deployment opportunities and the characteristics of the sea ice cover (almost total melt in summer, thinner sea ice and more divergent motion) means that buoys alone are not sufficient for characterizing Antarctic sea-ice drift on scales relevant for climate studies.

To supplement buoys in the Arctic, and to bridge gaps in coverage in the Southern Hemisphere, large-scale sea-ice drift data were derived from satellites imagery. The initial work by Ninnis et al. (1986) was followed by many investigators using a variety of satellite imaging sensor technologies as input, including visible and infrared radiometry (Emery et al., 1991), microwave radiometry and scatterometry (Agnew et al., 1997; Kwok et al., 1998; Liu et al., 1999; Lavergne et al., 2010; Girard-Ardhuin and Ezraty, 2012; Kimura et al., 2013), and synthetic aperture radar (SAR) (Kwok et al., 1990; Komarov and Barber, 2014; Muckenhuber et al., 2016). The number of investigators and studies introducing new sea-ice drift algorithms or data however does not translate into the availability of an abundance of multi-decadal, error-characterized climate data records of global sea-ice drift. Some datasets are not available, or not updated. Others cover only one hemisphere or are based on outdated algorithms. To our knowledge only two updated multi-decadal data records of sea-ice drift are available to the community: that of the National Snow and Ice Data Center (NSIDC) (Tschudi et al., 2020) and that of the French Institute for Marine Research (IFREMER) (Girard-Ardhuin and Ezraty, 2012). In this manuscript, we introduce a new 30-years Climate Data Record of year-round sea-ice drift, prepared and released by the EUMETSAT Ocean and Sea Ice Satellite Application Facility (OSI SAF).

The new CDR (OSI-455) is prepared using the Continuous Maximum Cross-Correlation (CMCC) method of Lavergne et al. (2010) applied on passive microwave imagery from the Special Sensor Microwave Imager (SSM/I), Special Sensor Microwave Imager Sounder (SSMIS), Advanced Microwave Scanning Radiometer for EOS (AMSR-E) and the Advanced Microwave Scanning Radiometer - 2 (AMSR2) satellite missions. Summer sea-ice drift fields are derived from a free-drift model tuned and forced with wind fields from the Copernicus Climate Change Service (C3S) atmosphere reanalysis ERA5 (Hersbach et al., 2020). The new CDR consists of daily product files with 24 h sea-ice drift vectors for the global sea-ice cover, and covers the period 1991-2020. It is readily available as OSI SAF (2022).

The manuscript is organized as follows: we first introduce the input satellite and reanalysis data used as input to prepare the CDR, and the on-ice buoy data used to validate it (Sect. 2). In Sect. 3 we document the algorithms implemented in the processing chains . We then describe the resulting CDR (Sect. 4.1) and validation results in the Arctic and Antarctic (Sect. 4.3). We finally present a short comparison to existing CDRs (Sect. 5), and discuss the merits and known limitations of our CDR (Sect. 6) before we conclude.

The present manuscript is prepared to be a detailed and citable reference to the new CDR. Users of the data are also referred to the extensive user documents: the Algorithm Theoretical Basis Document (Lavergne and Down, 2022a), the Product User's Manual (Lavergne and Down, 2022b), and the Scientific Validation Report (Lavergne and Down, 2022c). They are accessible from https://osi-saf.eumetsat.int/products/osi-455. In this manuscript, we use terms sea-ice *drift* and *motion* interchangeably to refer to the fact that sea-ice moves. We however make a distinction between sea-ice *displacement* vectors, *velocity* vectors, and sea-ice *speed* (among other terms) as explained in Sect. 4.2.

## 2 Input data

The CDR is mainly prepared from passive microwave satellite imagery, supported by wind vectors from atmosphere reanalysis data. It is validated using in situ buoy trajectories. Importantly, in situ data are only used for the validation, they do not enter the CDR itself, contrarily to Tschudi et al. (2020).

Table 1 summarizes the availability of the different input satellite imagery and wind data used to prepare the CDR.

### 2.1 Satellite imagery data

#### 2.1.1 U.S. DoD DMSP SSM/I and SSMIS

The SSM/I and SSMIS instruments on board the DMSP platforms of the US Department of Defence (DoD) have been the workhorse of sea-ice remote sensing for decades. Together with its predecessor the SMMR they allowed monitoring of the polar regions since the late 1970s.

They are conically-scanning multi-frequency microwave radiometers with imaging frequencies ranging from 19.3 GHz to near-90 GHz (85.5 GHz for SSM/I and 91.1 GHz for SSMIS). The two near-90 GHz channels (one at Vertical polarization, the other at Horizontal polarization) have the highest spatial resolution (13 x 14 km -3dB diameter of the Instantaneous Field Of View). We use brightness temperature measurements at these near-90 GHz channels (both polarizations) when preparing the CDR. The inclination of the orbit, and the width of the swath leave a so called polar observation hole where no sea ice imagery is available for motion tracking. This is poleward of $87°$ N for the SSM/I missions, and poleward of $89°$ N for the SSMIS missions.

The Scanning Multichannel Microwave Radiometer (SMMR, 1978-1987) had no near-90 GHz imagery and the very first SSM/I mission (F08, 1987-1988) experienced an early failure of its 85.5 GHz channels. They are thus not used in this first version of the CDR.

The Fundamental Climate Data Record (FCDR) Release 4 (10.5676/EUM_SAF_CM/FCDR_MWI/V004) from the CMSAF
is our source of SSM/I and SSMIS data (Fennig et al., 2020). It contains quality-controlled and intercalibrated Level-1B (swath-
based brightness temperature) data for SMMR and all SSM/I and SSMIS.

### 2.1.2 JAXA AMSR-E and GCOM-W1 AMSR2

The AMSR-E (2002-2011) and AMSR2 (2012 onwards) instruments are conically-scanning multi-frequency microwave ra-
diometers, with imaging frequencies ranging from 6.9 GHz to 89.0 GHz. The CDR uses the 36.5 GHz imagery (both polariza-
tions) as it offers the best compromise between retrieval accuracy and spatial resolution (7 x 12 km).

  The polar observation hole for the AMSR-E and AMSR2 missions is poleward of 89.5° N. For both missions, we access
brightness temperature data in swath projection (Level-1B data). We use the AMSR-E Level-1B data from the FCDR V003
(10.5067/AMSR-E/AE_L2A.003) of Ashcroft and Wentz. (2013). We access the GCOM-W1 AMSR2 Level-1B data directly
from the Japan Aerospace Exploration Agency (JAXA) G-portal.

### 2.2 Choice of microwave frequency

We thus select the near-90 GHz imagery channels for SSM/I and SSMIS, but the 36.5 GHz imagery channels for AMSR-E
and AMSR2. This choice is the result of a compromise between the level of details in the imagery (better with higher spatial
resolution, thus higher microwave frequency) and the stability of the imagery from one day to the next (better with channels
with less sensitivity to the atmosphere, thus lower microwave frequency).

  For the SSM/I and SSMIS mission, earlier investigations concluded that the near-90 GHz channels offer the best compromise
(Lavergne et al., 2010) as lower frequencies have much coarser resolution.

  For the AMSR-E and AMSR2 missions, the 36.5 GHz channels offer the best compromise. The 89 GHz channels have
higher resolution, but are also more affected by atmospheric liquid water path and surface melting. The 18.7 GHz channel has
previously been found useful for sea-ice motion tracking during summer (Kwok, 2008) but our experience preparing the CDR
is that the 36.5 GHz imagery provided at least as good results when compared to buoy trajectories (not shown).

### 2.3 Atmosphere reanalysis data

Wind fields from a Numerical Weather Prediction (NWP) model are used to drive a simplified (free-drift) model of sea-ice
motion during the summer season (Sect. 3.3). For this CDR, 10-m wind vectors are accessed from the C3S/ECMWF ERA5
reanalysis (Hersbach et al., 2020).

### 2.4 Auxiliary data

Land and open-water masks for each day in the CDR period are taken from the OSI SAF v2 Climate Data Records of sea-ice
concentration OSI-450 and OSI-430-b (Lavergne et al., 2019).

|  | Start date | End date | Significant gaps |
|---|---|---|---|
| ssmi-f10 | 1991-01-07 | 1997-11-14 | 1991-01-01 - 1991-01-06 |
|  |  |  | 1991-02-02 - 1991-02-08 (NH) |
|  |  |  | 1991-02-02 - 1991-02-12 (SH) |
|  |  |  | 1991-03-27 - 1991-04-18 |
|  |  |  | 1991-12-06 - 1991-12-12 |
| ssmi-f11 | 1992-01-01 | 1999-12-31 | 1996-05-30 - 1996-06-06 |
|  |  |  | 1997-02-19 - 1997-02-28 |
|  |  |  | 1997-03-19 - 1997-04-21 |
| ssmi-f13 | 1995-05-03 | 2008-12-31 |  |
| ssmi-f14 | 1997-05-07 | 2008-08-23 |  |
| ssmi-f15 | 2000-02-28 | 2006-07-31 |  |
| ssmis-f16 | 2005-11-20 | 2020-12-31 |  |
| ssmis-f17 | 2006-12-14 | 2020-12-31 | 2007-01-22 - 2007-02-02 (NH) |
|  |  |  | 2007-01-22 - 2007-02-04 (SH) |
|  |  |  | 2007-03-27 - 2007-04-05 |
|  |  |  | 2007-06-03 - 2008-03-26 |
| ssmis-f18 | 2010-03-08 | 2020-12-31 |  |
| amsr-aq | 2002-06-01 | 2011-10-04 | 2002-07-30 - 2002-08-08 |
|  |  |  | 2002-09-13 - 2002-09-20 |
|  |  |  | 2003-10-30 - 2003-11-06 |
| amsr2-gw1 | 2012-07-23 | 2020-12-31 |  |
| era5-wind | 1991-01-01 | 2020-12-31 |  |

**Table 1.** Availability (start, stop, and data gaps longer than 6 days) for the input satellite imagery and wind data used to prepare the sea-ice drift CDR.

## 2.5 In situ data for validation

We collect on-ice buoy trajectory data from a variety of sources. For Arctic sea-ice, we accessed the IABP archive of 3-hourly trajectories covering 1979 through 2016, and the archive of full-resolution (typically hourly) trajectories starting in 2008 and extending through 2020 (IABP, 2022). In addition, we access trajectories from the Ice Tethered Profilers (ITPs) (Toole et al., 2011), from Russian drifting stations NP-35 to NP-40, from the Tara schooner, the buoy array by Brümmer et al. (2011), the Ice Mass Balance Buoys (IMBs) from the Cold Regions Research & Engineering Laboratory (CRREL), and from GPS collars

in the Hudson Bay (Derocher, 2020). For Antarctic sea-ice we rely on two main data sources: the buoy trajectories compiled as part of the Atlas of Antarctic Sea Ice Motion (Schmitt et al., 2004) (1979-2000) and the buoys collected by the Alfred Wegener

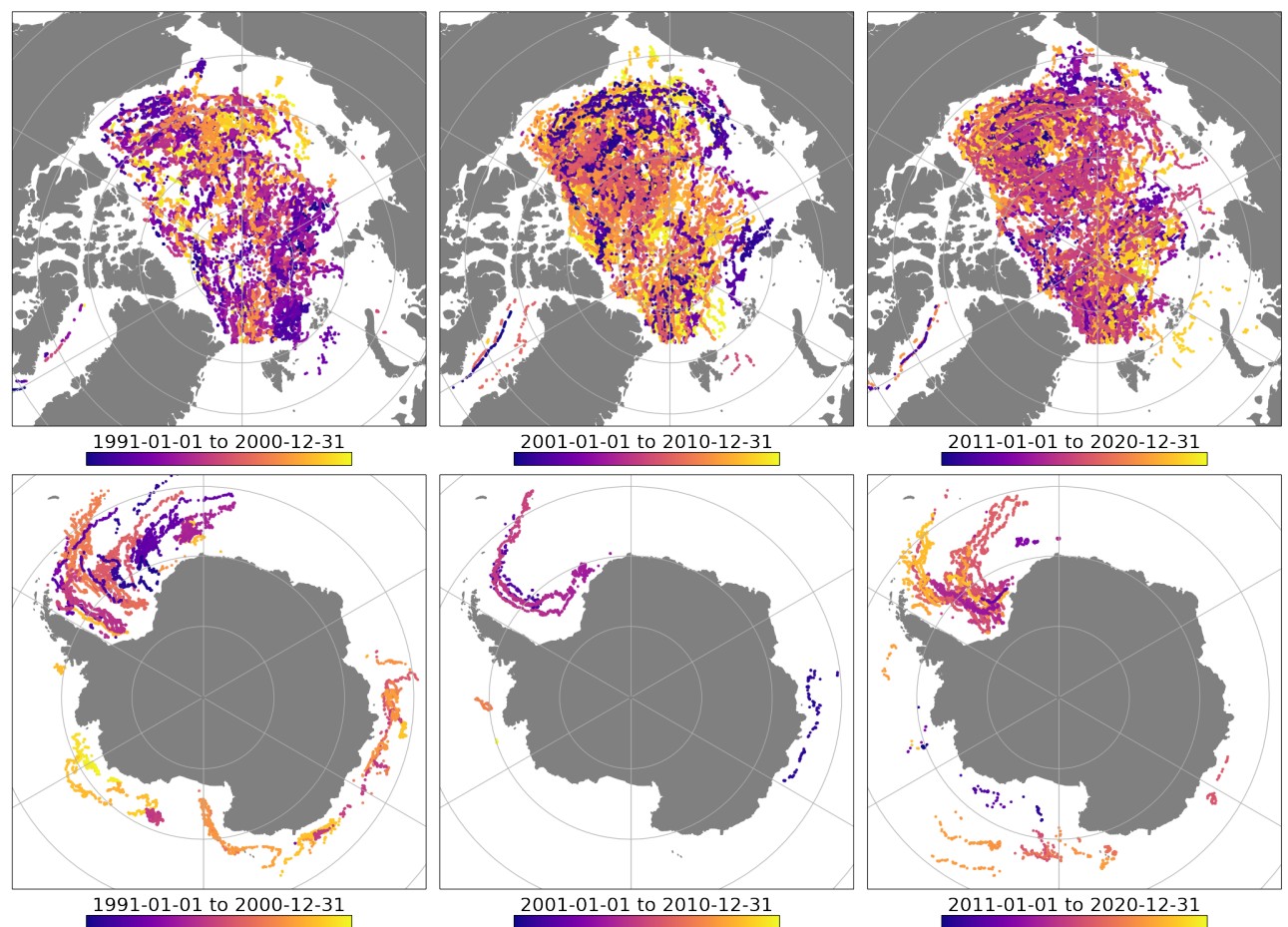

**Figure 1.** Maps of buoy trajectories available for validation across the three decades of the CDR (from left to right) and for both the Northern (top) and Southern Hemisphere (bottom). The color scale indicates the time within each decade.

Institute (AWI) and made available on the www.seaiceportal.de portal (Grosfeld et al., 2016). The collocation method we use (see Sect. 3.6) ensures that trajectories present in several collections are detected as duplicates and that only one instance enters the validation database.

Figure 1 shows the coverage of the buoy trajectories across the three decades of the sea-ice drift CDR. Notably, buoy positions plotted here are those that have been collocated with the CDR data files. This means that 1) only 1 buoy position a day is plotted, 2) no buoy data are plotted in the vicinity of the coastline (because there are no CDR data there), and 3) no buoy data are plotted in regions excluded from the main validation results of the CDR (e.g. Fram Strait and Hudson Bay) (see Sect. 6.3). Both summer and winter positions are plotted.

With this in mind, Fig. 1 gives an overview of the availability of buoy data in each hemisphere. We see an overall good coverage of the Arctic Ocean with buoys throughout the three decades (note less coverage in the peripheral Arctic seas such as

Baffin Bay, Laptev Sea, Kara Sea). For the Southern Hemisphere, there are much fewer data. The second decade (2001 - 2010) is when we have access to the least data despite having contacted a number of programmes and investigators. The buoys are more scattered in the various regions the Southern Hemisphere in the first decade of the CDR (1991-2000) than in the last one (2011-2020), when most of the buoy deployments are in the Weddell Sea.

## 3 Algorithms and processing details

To extract a CDR of sea-ice motion vectors from raw satellite imagery and atmosphere reanalysis fields require a number of steps and algorithms that are presented here. More details can be found in a dedicated Algorithm Theoretical Basis Document (ATBD) (Lavergne and Down, 2022a).

The methodology and source of sea-ice motion vectors depends on the season: in winter the vector fields are derived from satellite imagery, in summer they are processed from atmosphere reanalysis winds using a free-drift model. In the transition (autumn and spring) months, satellite- and wind-derived motion vectors are combined. These seasons and transition periods are defined in Table 2. The methodology to extract motion vectors from satellite imagery is very similar to that of the OSI SAF near-real-time sea-ice drift product (OSI-405) and introduced in Lavergne et al. (2010). Free-drift models have been used by a variety of authors to prepare year-round or summer-season drift data (Thomas, 1999; Tschudi et al., 2020; Brunette et al., 2022, among others).

| | Northern Hemisphere | Southern Hemisphere |
|---|---|---|
| Winter | Nov - Apr | Apr - Sept |
| Spring | May | Oct |
| Summer | Jun - Sept | Nov - Feb |
| Autumn | Oct | Mar |

**Table 2.** Seasons considered in the preparation and validation of the sea-ice motion CDR for the Northern and Southern Hemisphere. The calendar year is partitioned into (core) winter, (core) summer, and two transition months during spring and autumn.

## 3.1 Preprocessing of satellite and reanalysis data

Satellite imagery data are accessed as Level-1b data (calibrated brightness temperature $T_B$ data in satellite-swath projection) and prepared into daily averaged maps of $T_B$ with a 12.5 km spacing. This spacing roughly corresponds to the spatial resolution of the field-of-views for the imagery channels used in the processing (Sect. 2.1). We use the same approach as in Lavergne et al. (2010), in particular individual swath observations are weighted by their observation time (more weight towards 12 UTC than at 00 UTC and 24 UTC). This temporal weighting acts as a sharpening filter as it reduces the impact of the sub-daily sea-ice motion on the daily $T_B$ field. In addition to the daily gridded $T_B$, we compute the mean observation time for each

pixel in the gridded image using the same temporal weighting. This mean observation time can be useful for the users of the CDR to compute sea-ice velocity vectors Sect. 4.2 or to compute model-equivalent displacement vectors, and we use it when collocating the CDR with buoy data (Sect. 3.6). We thus do not follow the swath-to-swath approach of Lavergne et al. (2021) for the CDR, as it would have drastically increased the processing time, and since the benefits for the climate science community are not evident at this stage.

As in Ezraty et al. (2007), a Laplacian filter (second derivatives of the image intensity) is applied to the daily maps of $T_B$. The filter acts on two scales. First, it dampens large-scale intensity gradients across the images as well as intensity differences between the start and stop images as can be caused by passing weather systems. Second, it enhances small-scale intensity patterns in the image and stabilizes them in time, in view of their processing in the motion tracking algorithm.

In preparation for the free-drift model, the $u$ and $v$ components of the 10-m wind velocities are remapped and rotated onto the EASE2 grid of the CDR. The hourly wind components are then time-averaged across a 1 day period (12 UTC to 12 UTC), that corresponds to the motion vector duration of the CDR.

## 3.2 Winter sea-ice motion : satellite motion tracking

As in Lavergne et al. (2010), the motion tracking algorithm has two main components: the Continuous Maximum Cross Correlation (CMCC) method to detect motion vectors from pairs of (satellite) images, and the detection and correction of - so called - rogue vectors in the motion field. Here again, the interested reader is referred to Lavergne et al. (2010) and the ATBD (Lavergne and Down, 2022a) for detailed descriptions.

### 3.2.1 The continuous maximum cross-correlation method

The CMCC method is an evolution from the Maximum Cross-Correlation (MCC) algorithm. The latter has been widely used for motion extraction in geosciences (Emery et al., 1991; Girard-Ardhuin and Ezraty, 2012; Haarpaintner, 2006; Notarstefano et al., 2007; Schmetz et al., 1993, among others). The MCC is a block-based motion tracking algorithm from a pair of images. Blocks (also known as window, sub-image, etc...) from the first image hold a limited number of pixels (e.g. 7x7). One after the other, they are shifted by integer offsets (e.g. 2 image pixels in the x direction and -3 image pixels in the y direction) and the similarity with blocks in the second image is measured by the cross-correlation metric $\rho$. The offsets allowing the maximum cross-correlation is the solution drift vector. The MCC technique is robust and simple to implement, but its main drawback is the - so called - quantization noise because it results in quantized components of the drift vectors. This quantization noise is particularly an issue with rather coarse resolution images (12.5 km is our case) when the time duration between the two images is short (1 day in our case). In the past, sea-ice drift investigators have reduced the impact of the quantization noise by first refining the grid spacing of the images by bi-linear interpolation (e.g. by a factor of 3 or 5) before applying the MCC (Kwok et al., 1998; Tschudi et al., 2020). However this increases the size of the images and the computation time.

The CMCC is also a block-based motion extraction method, but the search for the best matching block is continuous over the two-dimensional plane of the image. Applying the CMCC requires on-the-fly computation of virtual image blocks, corresponding to a non-integer shift of the image pixels (e.g. by 1.82 pixels in the x direction, and -2.72 pixels in the y direction).

Once the method to compute the virtual image blocks is decided, it can be used in an iterative optimum-finding algorithm to search for the optimum shift vector that maximizes the cross-correlation metric $\rho$ between the virtual blocks from the first image and a block from the second image. As in Lavergne et al. (2010) we use bi-linear interpolation for computing the virtual blocks and a Nelder-Mead algorithm (Nelder and Mead, 1968; Lagarias et al., 1998) to optimize the cross-correlation metric. The CMCC thus generalizes the MCC with infinite refining of the image pixels, but without the need to store the images (on-the-fly computation). Although more complex to implement and, by nature, potentially less robust than the MCC technique, the CMCC has the distinct advantage of removing the quantization noise from the resulting vector field.

Importantly, we do not maximize the cross-correlation metric of individual brightness temperature images, but rather of the sum of the $\rho$ metrics over several $T_B$ images. Specifically, instead of running the CMCC separately on $T_B$ maps from the horizontal and vertical polarization channels and obtain two motion vector fields, we run a CMCC that searches for the maximum of the sum of the $\rho$ metrics over both polarization images, leading to a single vector field. The latter combines the information content of both polarization images. We adopt this strategy for each satellite mission separately, and not to combine several missions together (e.g. SSMIS F17 with AMSR2 GW1). This is mainly because the different missions have different orbits that observe the polar regions at different times during the day, thus not measuring the exact same sea-ice motion. Combining sea-ice motion fields from different missions is done at a later processing step, described in Sect. 3.4.

To constrain the search for the solution within a reasonable radius, the cross-correlation metric is multiplied by a sigmoid function that forces it to a bad score ($\rho = -1$) for displacement vectors corresponding to mean daily speeds in excess of 0.45 m.s$^{-1}$. This value is a compromise between allowing long enough drift distances, and keeping the computation time and the number of rogue vectors (see Sect. 3.2.2) low. In reality, ice floes can be recorded with higher hourly velocities in dynamic areas. However this sigmoid constrain corresponds to a sustained motion over 24 hours, and for a spatial extent of approximately 100x100 km. In Sect. 6.1, we discuss our choice of a maximum allowed speed in view of the validation results.

For this CDR, the CMCC is applied on an EASE2 grid with 75 km spacing, every 6$^{\text{th}}$ imagery pixel (itself on an EASE2 12.5 km grid). Each vector is built using sub-windows of 11 x 11 satellite imagery pixels. Vectors retrieved with a maximum cross-correlation of less than 0.3 are discarded and the product grid has a missing vector.

### 3.2.2 Detection and correction of rogue vectors

Once the motion tracking processing described above has been applied a first time across the sea-ice locations for a date in the CDR, a filtering step is taken to detect and - if possible - correct a number of obviously erroneous motion vectors. These are detected because they are much longer than neighbouring large-scale motion, often in diverging directions, hence the name rogue vectors. They can occur for several reasons, including convergence of the Nelder Mead algorithm in a local maximum, noise in the brightness temperature images, and edge effects in the image blocks.

As in Lavergne et al. (2010), the filtering step we implement is based on the distance from the end point of a displacement vector to the end point of the local average drift vector, computed from eight direct neighbour vectors. If this distance is less than a fixed threshold (10 km), the displacement vector is validated and the next one is tested upon. Otherwise, a new motion tracking optimisation is triggered with the CMCC. This time, the Nelder Mead algorithm is run with a constrain domain

centered on the local average drift vector, and with a smaller radius for the sigmoid (10 km). This effectively forces the Nelder Mead algorithm to converge to a local maximum compatible with the local average motion field. If this new optimization is successful, the corrected vector is entered in the product grid (with a specific flag). If no local maximum is found, or if it has a value less than $\rho = 0.5$, the new drift vector is discarded and the product grid has a missing vector. The filtering step starts from the rogue vector with the largest distance-to-average value, and continues until all rogue vectors are either corrected or declared impossible to correct. Each time a rogue vector is corrected, the local average drift field is updated, so that correcting rogue vectors has an immediate impact on the filtering of its neighbours. Our experience with the OSI SAF near-real-time product OSI-405 is that this is an effective way to not only detect, but also correct rogue vectors. The correction step allows for keeping the number of missing vectors low, and the correction via a second Nelder-Mead optimization yields more consistent results than more simple approaches like filling missing vectors with mean or median vectors. Our approach has some similarities to the correlation-relaxation method of Evans (2006) who kept track of several high-correlation vectors (not only the maximum) returned by the MCC and tested them against the local average vector field.

### 3.3 Summer sea-ice motion : wind-driven free-drift model

In the summer period, sea-ice motion tracking from passive microwave imagery satellite is much less reliable because of surface melting (e.g. melting of the snow cover, reduction of the difference in emissivity between first and multiyear sea-ice) and increased water content (liquid and vapour) in the atmosphere. This is particularly true when using the near-90 GHz imagery channels for SSM/I and SSMIS, which is the core of our satellite input data for the first decade of the CDR (Table 1). More recent missions like AMSR-E and AMSR2 offer lower microwave frequency channels at a better spatial resolution which allows some level of accuracy for motion tracking during summer but a) they are only available after 2002 and b) the resulting daily fields can have many gaps over melting sea-ice. To offer a consistent CDR over 30 years, we do not distribute the motion vectors retrieved from the AMSR-E and AMSR2 missions during summer, but rather use them to tune a free-drift model. We then use the free-drift model to prepare daily sea-ice drift fields during summer, that we distribute as part of the CDR.

The free-drift model is an alternative source of sea-ice motion vectors (Thorndike and Colony, 1982; Thomas, 1999; Brunette et al., 2022). It is a simplified theoretical model of sea-ice motion in which internal stresses are neglected (Leppäranta, 2011). Operating the model only requires a handful parameters that can be fixed or vary monthly, as well as daily wind velocity vectors as input. Because it neglects internal sea-ice stresses, the model is expected to be less valid in winter than in summer, and in the Arctic than in the Antarctic.

In the free-drift model, the sea-ice velocity vector $\boldsymbol{u}$ is expressed as:

$$\boldsymbol{u} = A\boldsymbol{U_a} + \boldsymbol{U_{wg}} + \boldsymbol{\epsilon} \quad \text{where} \quad A = |A|e^{-i\theta} \tag{1}$$

$\boldsymbol{U_a}$ is the 10 m wind velocity, $\boldsymbol{U_{wg}}$ is the under-ice ocean velocity, $\boldsymbol{\epsilon}$ is a residual term, and $A$ is the wind-ice–ocean transfer coefficient. $A$ can be expressed as a complex number. Its modulus $|A|$ is the scaling factor between wind and sea-ice speeds. Its argument is the turning angle $\theta$ between wind and sea-ice motion directions, due to the Coriolis effect.

Given a large enough set of collocated wind $U_a$ and sea-ice $u$ velocity vectors, one can estimate the free-drift parameters $|A|$, $\theta$, $U_{wg}$ and the residual $\epsilon$. This is the step of tuning the free-drift model (Sect. 3.3.1).

Given the parameters of the free-drift model $|A|$, $\theta$, and $U_{wg}$ and 10-m wind velocity vectors $U_a$, one can compute sea-ice drift velocity vectors. This is the step of running the free-drift model (Sect. 3.3.2).

The free-drift equation Eq. (1) derives from the general momentum balance equation under specific conditions, including that internal sea-ice stresses are negligible (Leppäranta, 2011). Strictly speaking, this can only happen if the sea-ice concentration is rather low, e.g. below 0.8. However, we apply the free-drift model to conditions with much larger concentrations in this CDR. This is because we use the free-drift model only as a parametric formula for sea-ice motion, and we tune the parameters $|A|$, $\theta$, and $U_{wg}$ against drift data. The values for these parameters are not derived analytically from the free-drift equation theory (involving the Nansen and Rossby number, the Coriolis constant, etc). Because we only use the free-drift equation as a model to fit data, the assumption is that our tuned parameters compensate to the best of their capacity for the theoretical limitations of the free-drift model.

### 3.3.1 Tuning the free-drift model

We tune the free-drift model against maps of sea-ice motion vectors obtained with the CMCC algorithm from the 36.5 GHz imagery of the AMSR-E and AMSR2 missions (2002-2020), as well as ERA wind data projected and rotated onto the CDR grid. Parameter-tuning is performed on a monthly basis, using all available years (2002-2020). We thus obtain 12 monthly maps of parameters $|A|$, $\theta$, and $U_{wg}$. This is a similar approach to Thomas (1999) who found that monthly parameters allowed better accuracy of the modelled wind-driven motion field. For implementation reasons, we use a least-square solver with complex numbers, their real part being the x-component of the vectors and the imaginary part being the y-component (Mozo, 2017).

A gap-filling by extrapolation is then applied on the monthly maps of parameters. This extrapolation is necessary because we tune the wind-driven model on the later part of the timeseries (AMSR-E and AMSR2) but we must be able to apply the model in the early decade of the CDR as well, when sea ice covered more area (in the Arctic). We underline that the extrapolation is performed after the parameter tuning and cannot degrade its performance. Finally, the maps of parameters are smoothed using a Gaussian weighting filter with a 62.5 km sigma. The monthly maps of parameters (and residuals from the least-square fitting) are inspected visually, and finally saved in netCDF files to later run the free-drift model.

Figure 2 shows the retrieved parameters for Northern Hemisphere sea-ice motion for the month of July. These maps confirm what other investigators reported, e.g. that the wind–ice–ocean transfer coefficient $|A|$ (panel a) is around 2% and the turning angle $\theta$ (panel b) is around $-30°$. $|A|$ takes smaller values North for Greenland and the Canadian Archipelago where older and thicker sea ice with larger internal stresses is. This is expected from the derivation of the free-drift model (Leppäranta, 2011) and similar to results obtained by Brunette et al. (2022) and that prompted their parametrization of $|A|$ on sea-ice thickness from reanalysis data (which we do not adopt in this CDR). The spatial gradient of $|A|$ shown on Fig. 2 is less pronounced than that of (Brunette et al., 2022, Fig. 4) because we show data for a summer month while they show year-round data. Figure 2 (panel c) is a map of the retrieved under-ice ocean currents that exhibits well known circulation patterns such as the Beaufort Gyre and the Transpolar drift, all the way from the Laptev Sea and extending into Fram Strait and East Greenland Current.

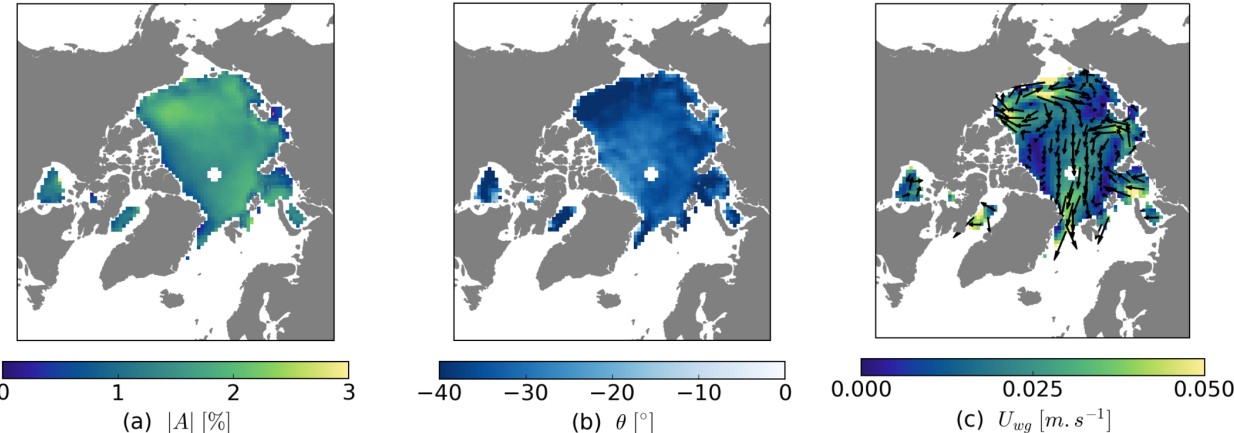

**Figure 2.** Maps of retrieved free-drift parameters for Northern Hemisphere sea-ice motion for the month of July: (a) wind–ice–ocean transfer coefficient $|A|$, (b) turning angle $\theta$, and (c) under-ice ocean currents $\boldsymbol{U_{wg}}$. These maps show the parameters before the spatial extrapolation. The white pixels around the North Pole and along the coasts are filled in the extrapolation step. On panel (c), the vector field was thinned to improve readability.

Maps of retrieved parameters for the Southern Hemisphere (Fig. 3) for the month of December confirm that the wind–ice–ocean transfer coefficient $|A|$ (panel a) is larger in the Antarctic than in the Arctic (compare to Fig. 2 panel a), which is compatible with a generally thinner and less compact sea-ice cover. The turning angle $\theta$ (panel b) takes on positive values around $+30°$, which is expected from the Coriolis effects. $|A|$ generally takes lower values close to the continent and the Antarctic Peninsula in the Weddell Sea, which is compatible with thicker sea ice and a larger impact of neglected internal stresses. The map of under-ice currents in panel (c) reveals two well known features: the Weddell Sea gyre and especially the slope current running northward and parallel to the Antarctic Peninsula, and the Ross Sea gyre. Note that only the high-latitude part of these currents are revealed because this is where the free-drift model was tuned in December.

Table 3 reports the mean free-drift parameter values $|A|$ and $\theta$ as well as the magnitude of the residuals $\epsilon$ by months, in both hemispheres. It provides a summary of the free-drift parameters for comparison with other investigators, but we only use the monthly parameter maps in the CDR, and mainly for the spring-summer-autumn months. It also shows that parameters $|A|$ and $\theta$ have a pronounced seasonal cycle which justifies a tuning strategy by month (Thomas, 1999).

All in all, this short analysis of the obtained monthly parameter maps give us confidence that they can be used in generating part of the CDR, as a gap-filler for satellite-based motion vectors during the summer season. We discuss the limitations of this approach in Sect. 6.

### 3.3.2 Running the free-drift model

We obtain daily maps of wind-driven sea-ice velocity vectors for each day of the CDR by entering the rotated and gridded ERA5 daily wind velocity vectors as input to Eq. (1), along with the corresponding monthly parameters maps. Since both the

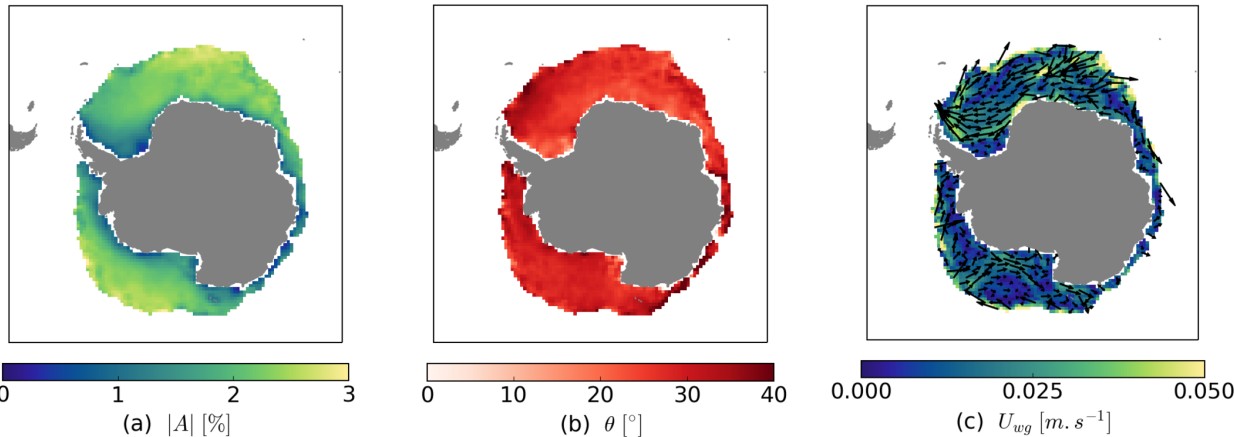

**Figure 3.** Same as Fig. 2 but for Southern Hemisphere sea-ice motion and for the month of December.

| | Northern Hemisphere | | | Southern Hemisphere | | |
|---|---|---|---|---|---|---|
| Month | $\langle |A| \rangle$ / % | $\langle \theta \rangle$ / degree | $\langle U_{wg} \rangle$ / ms$^{-1}$ | $\langle |A| \rangle$ / % | $\langle \theta \rangle$ / degree | $\langle U_{wg} \rangle$ / ms$^{-1}$ |
| Jan | +1.5 | -18.2 | +0.022 | +1.8 | +31.3 | +0.017 |
| Feb | +1.5 | -17.6 | +0.022 | +1.5 | +28.8 | +0.022 |
| Mar | +1.4 | -17.5 | +0.022 | +1.8 | +23.6 | +0.020 |
| Apr | +1.4 | -18.3 | +0.020 | +1.9 | +21.1 | +0.020 |
| May | +1.4 | -23.3 | +0.021 | +2.0 | +20.3 | +0.020 |
| Jun | +1.4 | -27.8 | +0.020 | +2.0 | +19.9 | +0.021 |
| Jul | +1.5 | -34.3 | +0.020 | +2.1 | +20.2 | +0.023 |
| Aug | +1.6 | -35.6 | +0.018 | +2.1 | +20.7 | +0.023 |
| Sep | +1.7 | -30.8 | +0.017 | +2.1 | +21.6 | +0.023 |
| Oct | +1.8 | -24.5 | +0.020 | +2.1 | +23.0 | +0.023 |
| Nov | +1.8 | -21.0 | +0.022 | +2.1 | +25.3 | +0.022 |
| Dec | +1.7 | -19.1 | +0.022 | +2.0 | +28.1 | +0.018 |

**Table 3.** Average free-drift parameters $|A|$, $\theta$, and under-ice current speed $U_{wg}$ per month for the Northern Hemisphere (left) and Southern Hemisphere (right).

ERA5 winds and the parameter files are prepared on the EASE2 grid with 75 km spacing, the resulting wind-driven vectors are directly on the same grid as the satellite-derived vectors. To ensure a smooth transition across the months, we always compute two sea-ice velocity fields based on the parameter files from the two "bracketing" months. These two velocity fields

are then blended by linear interpolation, weighted by the difference between the simulation date and the mid-month dates of the parameter files. For example, for a free-drift simulation for June 21st, the two bracketing months are June (16th) and July (16th). Sea-ice velocity vectors are finally converted to sea-ice displacement vectors by mutiplying the components by 24h (see a note about this conversion to displacement in Sect. 4.2).

A last step in the wind-drift calculation is to apply the sea-ice mask for the current date to ensure that wind-driven vectors are only kept over sea ice. This mask is from the EUMETSAT OSI SAF Sea Ice Concentration CDR v2 (see Sect. 2.4).

### 3.4 Multi-source merging

The main field of our sea-ice motion CDR is a multi-source product with daily-complete maps of motion vectors throughout the 30 years period, everyday and in all seasons. This requires to combine and sometimes gap-fill the single-sensor products based on satellite imagery and the wind-driven motion fields from the free-drift model. These wind-driven fields 1) fully replace the single-sensor satellite products in the summer season, 2) are blended with the single-sensor satellite data in the spring and autumn transition months, and 3) gapfill whole winter days when satellite drift data are fully lacking (e.g. no satellite imagery data). The strategies for this multi-source merging are presented below.

The merging algorithm has three main steps for each date in the data record. We first select what single-sensor and wind products are available to contribute to the merged product. We then optimally merge the selected sources at each grid locations. Finally, possible remaining gaps in the vector field are interpolated from neighbour vectors to obtain daily-complete maps.

#### 3.4.1 Selection of single-sensor and wind drift vectors

First, a check is made that there are enough vectors within each single-sensor satellite product. This is done calculating the ratio between the number of available motion vectors and the number of possible sea-ice grid cells, away from a wide polar observation hole (latitudes poleward of 86 N), to ensure that the check is consistent across satellite missions (with varying widths of the polar observation hole, Sect. 2.1). A threshold is set at 40%, below which the entire single-sensor motion field is discarded as not having sufficient data. This occurs more often at the start of the CDR period, where there are only one or two satellite missions and some days with missing data (Table 1).

#### 3.4.2 Optimal merging of satellite-based motion vectors

Here, the terminology *optimal* relates to the use of uncertainty estimates in terms of variance, as weights in the merging formula. Let $(\mathbf{d}X, \mathbf{d}Y)_1$, $(\mathbf{d}X, \mathbf{d}Y)_2$,... be the $S$ available single-sensor motion vectors at a given grid location (typically $S$ is between 2 - 4 in the winter season thanks to the overlap of satellite missions, see Table 1). The multi-sensor drift vector $(\mathbf{d}X, \mathbf{d}Y)_m$ at this same location is computed as:

$$(\mathbf{d}X, \mathbf{d}Y)_m = \frac{\sum_{k=1}^{S}(\mathbf{d}X, \mathbf{d}Y)_k \times \frac{1}{(\sigma_k^{12utc})^2}}{\sum_{k=1}^{S} \frac{1}{(\sigma_k^{12utc})^2}} \tag{2}$$

$\sigma_k^{12utc}$ are the uncertainties for the vector components for source $k$, adjusted to a 12 UTC to 12 UTC drift period (see Sect. 3.5).

Equation (2) is a simplification of the full equation for combining several multi-dimensional Gaussian estimates into an optimal (*aka* Maximum Likelihood) estimate. The simplifications are:

1. Both $\mathbf{d}X$ and $\mathbf{d}Y$ components use the same value of $\sigma_k^{12utc}$;

2. We do not consider correlation between the uncertainties of $\mathbf{d}X$ and $\mathbf{d}Y$ components;

3. We do not consider correlation between the uncertainties of neighbouring vectors in the single-sensor products;

4. We do not consider correlation between the uncertainties of the $S$ single-sensor products;

The simplifications above are used as pragmatic solution to decrease computation time and in absence of estimates for these correlations. The uncertainties we use as weights in Eq. (2) are those documented in Sect. 3.5 and are based on the statistical results from the validation against buoy data (Sect. 4.3).

Applying Eq. (2) at all sea ice grid locations with at least one single-sensor drift vector provides a new map of multi-sensor vectors $(\mathbf{d}X, \mathbf{d}Y)_m$. This new map might still have data gaps, which are filled by spatial interpolation, as described in the next section.

### 3.4.3 Merging with wind-based motion fields

Generally, the satellite-based single-sensor vectors are the only input to the merging during the winter season, and the wind-driven drift vectors the only input in the summer season.

In the transition months (spring and autumn, Table 2), both single-sensor satellite and wind-driven drift products contribute, weighted by their uncertainties. The uncertainties are modified taking into account the day in the transition month as follows. In the spring month, the uncertainties of the CMCC-calculated drift vectors are at their nominal values (typically 2.5 to 3.5 km standard deviation in the Northern Hemisphere, 3.5 to 4.5 km in the Southern Hemisphere, see Sect. 4.3) at the start of the month, and at a high value (10 km standard deviation) at the end of the month. The uncertainties on each spring day are linearly interpolated between these end points. For the wind drift, the reverse is true - the values are set to high (10 km standard deviation) at the start of spring month and are nominal at the end. This ensures that the wind-driven motion fields are smoothly introduced into the multi-source fields of the CDR. The same applies in the autumn transition month, with the uncertainties of the satellite and wind-derived motion fields ramping in the opposite direction.

Wind-derived motion fields have sometimes to be used during the winter season, when missing satellite imagery leads to a lack of satellite motion vectors. The wind-derived motion field is then used directly in the multi-source product. The dates at which this occurs are listed in Appendix A, and a specific `status_flag` value is used in the product files. Because we aim at a gap-free CDR, we thus apply the free-drift model in winter, despite it being theoretically less accurate because of larger internal stresses. Given that sea-ice motion is very variable from day to day, mostly driven by wind forcing, we deemed it makes more sense to use the free-drift model than other interpolation methods (e.g. a linear interpolation from neighbouring

days, or reverting to a climatology). We underline that winter free-drift estimates are only to fill whole days at a time (not grid cells here or there which are interpolated from neighbouring vectors). No smooth transition is implemented from the day before or after to not degrade high-quality satellite-based sea-ice drift vectors with lower quality winter free-drift vectors: this is an abrupt replacement. Users who do not want to use free-drift estimates during winter are invited to use the `status_flag`.

### 3.4.4 Spatial interpolation for gap filling

Once the satellite and wind-derived drift vectors have been combined, a gap-filling runs. The gap filling is handled with spatial interpolation.

The interpolation weight is function of the distance to neighbouring grid cells, is gaussian-shaped, with a reference length of 200 km (standard deviation). The maximum radius for finding neighbours is set to 300 km (for reference, the grid spacing between vectors is 75 km).

We do not fill gaps towards grid cells where the single-sensor satellite products would never contain a motion vector, such as in straits, along coastlines and in peripheral closed seas

At the end of these three steps, we obtain a multi-source gap-free year-round sea-ice motion field that is the core of the OSI SAF CDR, and we also distribute all single-sensor and wind-driven motion fields for interested users. The multi-source CDR is sometimes referred to as the multi-oi product because it is the result of an optimal merging and interpolation from multiple sources.

## 3.5 Uncertainties for each vector

We provide an uncertainty estimate for each motion vector in the CDR, both for the satellite-based, wind-derived, and multi-source products. We base our uncertainty values of the satellite-based and wind-derived products on their aggregated validation statistics (specifically the RMSE) against buoy trajectories. Validation statistics are binned by category of status flags, by hemisphere, and by type of input source (the SSM/Is, the AMSRs, and wind-derived products defining 3 broad categories). These tabulated RMSEs are then used to assign an uncertainty value (1-$\sigma$) to each motion vector. The same value (noted $\sigma_k$) is used for both the $\mathbf{d}X$ and the $\mathbf{d}Y$ components of the motion vector because the validation against buoy trajectories does not reveal any significant difference between the two components.

The uncertainty values $\sigma_k$ introduced above are valid when using the single-sensor ice drift vectors with their associated maps of start and end time provided in the product files. As shown later, these can easily vary between 8 UTC and 16 UTC across the product grid depending on the orbit and instrument characteristics (see Fig. 4). When rather using the vectors as if they were from 12 UTC and 12 UTC, the uncertainties must be raised. We compute the raised uncertainty $\sigma_k^{12utc}$ with a 2nd order polynomial formula:

$$\sigma_k^{12}(\delta_t) = a \times \delta_t^2 + b \times \delta_t + \sigma_k \tag{3}$$

where $\delta_t = |t - 12\ UTC|$ has units hours. The values of coefficients $a = 0.015$ and $b = -0.005$ were obtained by running a validation experiment where we deliberately collocated buoys and satellite product with wrong time information, allowing us to explore the increase in uncertainties with an increase in time mismatch (from -10 hours to +10 hours). This exercise was conducted with the near-real-time OSI SAF sea-ice drift product, and was not repeated for the CDR as we believe the coefficients apply.

We compute maps of $\sigma_k^{12utc}$ using Equation (3) for the single-sensor products that are input to the multi-sensor merging step (see Equation (2) in Sect. 3.4). The uncertainty values of the merged multi-source product are computed by combining the uncertainty values of the single-sensor products $\sigma_k^{12utc}$.

### 3.6 Validation against buoy trajectories

#### 3.6.1 Reformatting and quality-control of buoy trajectories

On-ice buoy trajectories are accessed in a variety of (text-based) formats and reformatted to netCDF files that follow the Climate and Forecast convention (CF-community, 2022). In this process, a set of tests are applied to each portion of the drift trajectories to detect and discard erroneous positions that would lead to degraded statistics. In addition, visual inspection of each buoy trajectory led to black-listing a number of buoys.

The automatic quality-control steps are:

– reorder the position records chronologically;

– remove duplicate position records (same lat/lon/time);

– remove portions of trajectories where the position flickers between fixed positions;

– remove position records between which the speed is larger than three times the standard deviation from the mean speed, where the mean speed is computed over the whole trajectory. This is to detect and remove rogue locations along the trajectory. This test would possibly be more effective with a running window along the trajectory, which will be considered in future versions of the CDR.

#### 3.6.2 Collocation strategy

In order to compare the sea-ice motion vectors from the CDR (noted *prod*) with the buoy trajectories (noted *ref*), they need to be collocated together. Collocation is the action of selecting or transforming one or both data sources so that they represent the same quantity, at the same time and at the same geographical location.

Because the OSI SAF sea-ice motion product comes with two flavours of time information, two collocation strategies are defined:

– One using a 2D collocation, in which the sea-ice motion is considered representing a displacement from day 12 UTC to 12 UTC the following day;

– One with a 3D collocation, in which the position-dependent start and end times of the motion vectors (corresponding to
the mean observation times computed in Sect. 3.1) are used.

In the validation of the sea-ice drift CDR, the single-sensor products are collocated with a 3D collocation strategy, the wind-driven drift product is collocated with a 2D collocation strategy (since the wind-driven drift is simulated from 12 UTC to 12 UTC), and the multi-source product is collocated with a 2D collocation (since it is considered valid from 12 UTC to 12 UTC). The 2D and 3D collocation strategies differ by how they handle the temporal collocation, but both use a nearest-
neighbour strategy for the spatial collocation. The temporal collocation impacts how position records along the buoy trajectories are selected. In the 2D collocation, the start (resp. end) records for the buoy drift vector are those closest to 12 UTC (resp. 12 UTC the following day). In the 3D collocation, the local start (resp. end) records for the buoy drift vector are those closest to the local (space-varying) mean observation time of the start (resp. end) image entering the motion tracking step.

The following criteria apply for a matchup pair (*ref*, *prod*) to enter the matchup database:

1. the distance from the start position of the buoy vector to the start position of the product vector must be smaller than 40 km;

2. the difference in start time between the buoy and the product vectors must be less than 3 hours;

3. the difference in duration between the reference and the product vectors must be less than 1 hour;

4. A buoy vector must be surrounded by four valid product vectors. Although only the nearest of these four is considered in
the spatial collocation, this constraint is introduced to exclude validation data in the outer edges of the vector field, like in the marginal ice zone or in coastal regions (land-fast ice). Since some of the ice-tethered buoys are designed to continue floating when the sea ice floe they are attached to melts, this constraint is also an effective way for not collocating ocean drift measurements with our sea-ice drift CDR.

5. For a given date, any two buoy vectors must be separated by at least $3 \times 75.0 = 225$ km. This constraint is introduced
so that all data pairs entering the matchup database are independent from each others. Also, it ensures that buoys that are reported by several providers (e.g. an ITP reported by IABP, see Sect. 2.5) are not counted twice in the validation statistics. The same applies for small buoy arrays that would otherwise have more weight than individual buoys.

## 4   The resulting data record and its evaluation against buoy trajectories

### 4.1   A global data record of sea-ice motion covering 1991-2020

The resulting data record is the EUMETSAT OSI SAF global climate data record of sea-ice drift version 1 (OSI-455), released in November 2022.

The CDR covers the 30 years period from January 1991 through December 2020. It consists both of the merged multi-source product files, and of the individual single-sensor and wind-driven product files. The merged product files are considered the

entry point for the CDR, while the single-sensor and wind-driven files are made available for traceability and more advanced
users that can cope with data gaps. The satellite single-sensor product files are only available for the winter (and spring and fall) season, while the wind-driven product files are only available for the summer (and fall and spring) season (see Table 2 for the definition of these terms in the context of the CDR).

Although buoys are a trusted source of sea-ice motion information and can capture high-resolution and high-frequency patterns that are not accessible from satellite missions we use, we do not include them as input to the merged multi-source
product. This is conversely to what is done for the NSIDC sea-ice motion dataset, where buoys are used in the Arctic (Tschudi et al., 2020). Our main motivation is to keep buoys as an independent source of validation data. Second, buoys are point-like observations while our satellite-based vectors represent the average motion of large areas. Combining the two scales in the merging was not straightforward. Although it improves in version 4 of the NSIDC data record, the inclusion of buoys has been shown to lead to non-physical discontinuities in the motion fields of earlier versions (Szanyi et al., 2016; Tschudi et al., 2020).

Each product file holds maps of 24 h motion vectors for either the Northern or Southern Hemisphere. They are formatted as NetCDF files following the Climate and Forecast (CF, CF-community (2022)) convention. In addition to the sea-ice drift information and associated uncertainties, they include geolocation information and status flags.

In each NetCDF file, the variables are : `time`, `time_bnds`, `t0`, `t1`, `lat`, `lon`, `xc`, `yc`, `Lambert_Azimuthal_Equal_Area`, `lat1`, `lon1`, `dX`, `dY`, `uncert_dX_and_dY`, and `status_flag`. We provide a brief description of each of these below.

Variables `time` (a scalar), `time_bnds` (two scalars), `t0` (map) and `t1` (map) provide time information for the motion field. `time_bnds` holds the start and end date (12 UTC) of the displacement vectors. The valid time for the product, `time`, is arbitrarily set to the end date of the displacement. `t0` (resp. `t1`) are 2D maps of the start (resp. end) timestamp of each motion vector in the product file. For the wind-driven and multi-source products, it is 12 UTC for all vectors. For single-sensor drift products, it corresponds to the mean observation time of the start (resp. end) satellite image. Figure 4 shows the start time, end
time, and drift duration (`t1` minus `t0`) for a Northern and Southern Hemispheres motion field based on the AMSR2 mission. Note how the orbit pattern of the AMSR2 mission translates into space-varying fields of the start and end time (not 12 UTC everywhere), and how these lead to drift durations that are not exactly 24 h across the product grid.

Variables `lat`, `lon`, `xc`, `yc`, and `Lambert_Azimuthal_Equal_Area` together define the geolocation information of the product file. `xc` and `yc` (both with units $km$) are the grid coordinates of the product in the polar EASE2 grid defined by the
`Lambert_Azimuthal_Equal_Area` variable, while `lat` and `lon` are the corresponding latitudes and longitudes. The spacing in both `xc` and `yc` is 75 km, the grid spacing of the CDR product.

Variables `lat1`, `lon1`, `dX`, and `dY` define the drift vectors. `dX` and `dY` are the components (units $km$) of the drift vectors along the two axes of the grid (`xc` and `yc`). `lat1` and `lon1` are the coordinates of the end point of the drift vectors. To provide vector components in two coordinate systems is redundant, but we keep both and let users adopt either. Our preferred way to
use the CDR is with `dX` and `dY`. The sign convention of `dX` and `dY` is critical to understand. `dX` is positive for a motion from left to right in the product grid, while `dY` is negative for a motion from top to bottom in the product grid. This is because of the orientation of the grid axes `xc` and `yc`. The sign of `dX` and `dY` is illustrated on Fig. 5.

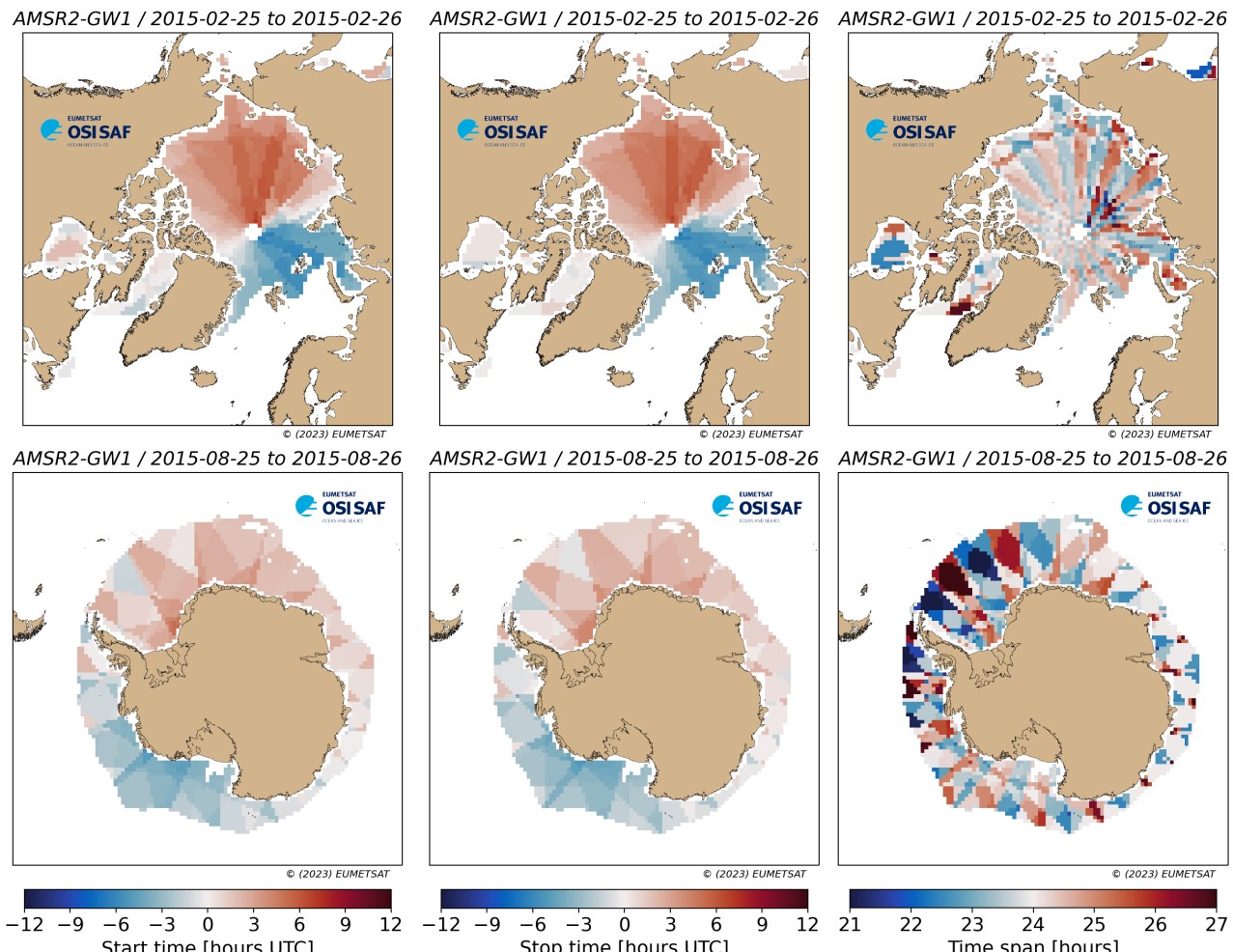

**Figure 4.** Example fields of start time `t0` (left), end time `t1` (middle), and drift duration (`t1` minus `t0`) (right) for a Northern (top) and Southern (bottom) Hemisphere sea-ice motion field from the AMSR2 mission. The start and end time are shown as a time-difference to 12 UTC.

Variable `uncert_dX_and_dY` holds fields of retrieval uncertainties. These are 2D fields of uncertainties that vary on a daily basis as a result of the processing steps, both for the single-sensor and the merged product (Sect. 3.5). A single uncertainty field is provided in each product file for both the $\mathbf{d}X$ and $\mathbf{d}Y$ components (hence the name). This is because our method to derive the uncertainties from the statistics of the validation exercise does not support providing different values for the two components at this stage. The uncertainty in this variable is to be interpreted as 1-sigma of the uncertainty distribution in both $\mathbf{d}X$ and $\mathbf{d}Y$.

Finally `status_flag` holds processing and status flags to indicate why a grid cell does not hold a motion vector (rejection flag) or the a-priori quality of a motion vector (quality flag). Variable `status_flag` takes values from 0 (missing satellite

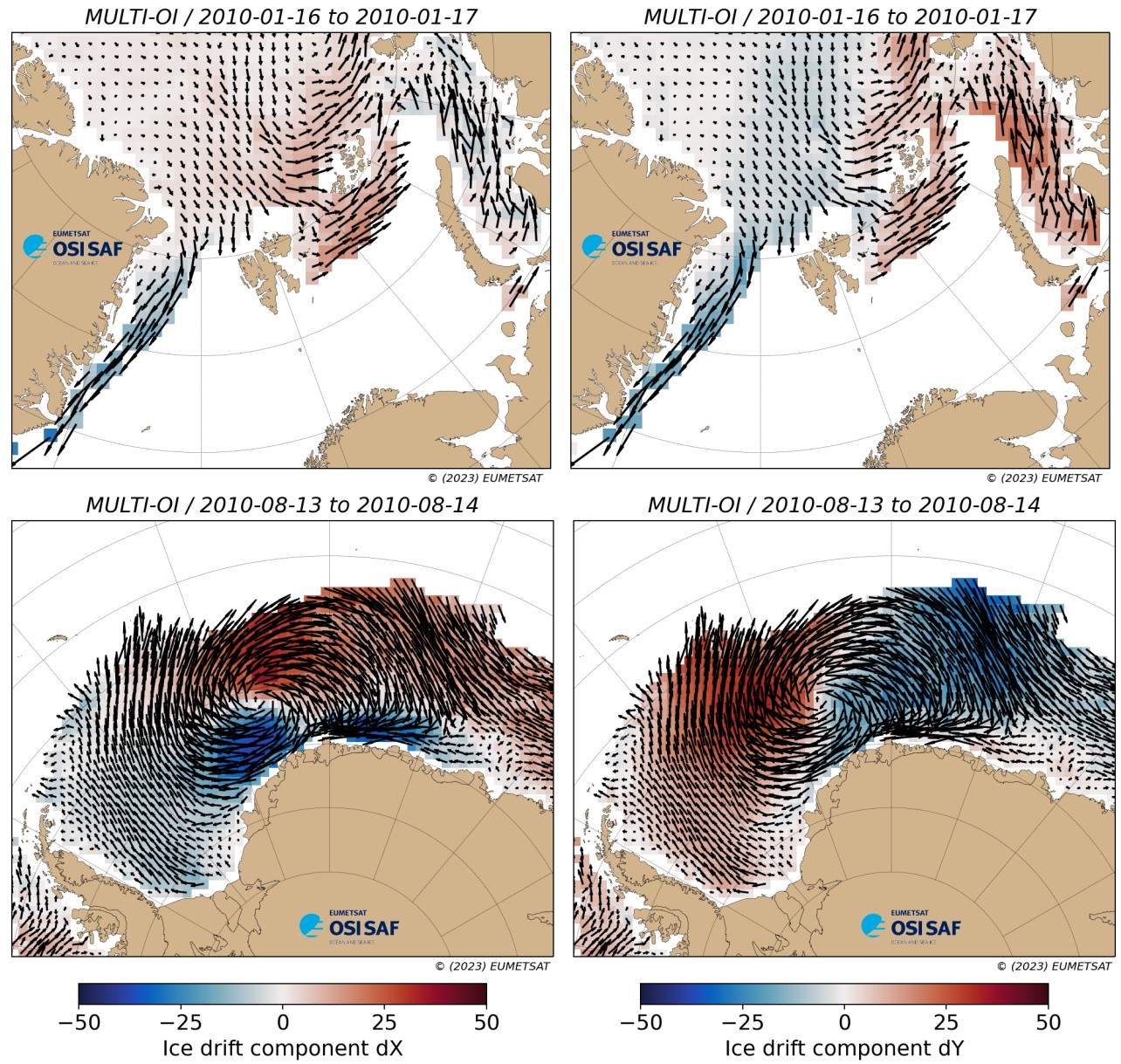

**Figure 5.** Example fields of `dX` (left) and `dY` (right) for a Northern (top) and Southern (bottom) Hemisphere sea-ice motion field from the multi-source product files. Note the sign convention for `dY`.

imagery data) to 30 (nominal quality). Values 0 to 19 are rejection flags, values 20 to 30 are quality flags. Each time the motion vector in the product does not originate directly from the nominal CMCC algorithm applied to satellite imagery, a flag lower than 30 is reported. For example, value 25 indicates that a motion vector originates from the blending of satellite and

wind-driven vector. All flag values are defined in the product file (following the CF convention) and in the Product User's Manual (Lavergne and Down, 2022b).

## 4.2 Vectorial representation of sea-ice drift

There are many options for reporting and storing sea-ice motion vectors in gridded product files. The lack of a common practice can rapidly become an impediment to the adoption of products, because users have to understand how each product is stored, and potentially how to transform it back to something they can process. It can also lead to confusion and misinterpretation when quantities that are not comparable are compared. We do not seek to impose or even propose a common practice here, but to explain and justify the vectorial representation we selected for the OSI SAF CDR.

We first observe why these many options and the confusion. Firstly, and unlike many modeled and observed geophysical parameters, sea-ice motion is a vector quantity: it requires a pair of scalars to be fully described. There are many choices for these scalars: component of the vector along the grid of the product (our $\mathbf{d}X$, $\mathbf{d}Y$), component of the vector along standard directions, e.g. latitudinal (*aka* south-north or meridional) and longitudinal (*aka* west-east or zonal) components, components of the vector in a polar coordinate system, i.e. magnitude and direction, or even two pairs of latitude and longitudes: the geographical position at the start and end of displacement (our ($\mathtt{lat_0}$, $\mathtt{lon_0}$), ($\mathtt{lat_1}$, $\mathtt{lon_1}$), but also the position records in a buoy trajectory), to name the most common. Secondly, sea-ice motion can both be represented as a time intensive or a time extensive quantity. All satellite-based products retrieved from pairs of images are time extensive: they measure the displacement between the time of the two images. The motion between any two position records along a buoy trajectory is also a time extensive quantity: a displacement. Conversely, sea-ice motion in the equations approximated by a geophysical model is a time intensive quantity: the sea-ice velocity. Velocity can be integrated in time (in a Lagrangian sense) to yield a displacement. In theory, a displacement cannot be turned into a velocity without making an hypothesis on the motion between the start and end times of the displacement. In practice however, displacements are very often expressed as if they were velocities. For example, the component of a displacement with units [m] will be reported with units [m.s$^{-1}$] by dividing the displacement by the time duration between its start and end times. This is one source of confusion. Another source of confusion is that velocities are often averaged over time periods, and this average is performed in a Eulerian sense (average of the velocities at a given location, over a time period). Geophysical models typically store hourly or daily averaged velocities, from which weekly or monthly velocities are computed. A weekly (Eulerian) averaged velocity and a weekly (Lagrangian) integrated displacement expressed as a velocity are not the same quantities, although the mismatch can be small if the velocity fields are spatially uniform across distances larger than the total displacement. We argue that the most correct way of matching model velocities to satellite displacement vectors is to integrate the model velocities into trajectories. By the same token, the displacement vector between two position records of a buoy, not the average of the point-to-point velocity vectors along the trajectory, is the closest quantity to a satellite displacement vector.

The above explains many of the choices we made when designing what variables to write in the product files, and how to conduct and report the validation against buoys. We use polar EASE2 projections both for the satellite imagery (the 12.5 km spacing image grid) and the start location of our vectors (the 75 km spacing product grid). We prepare daily averaged maps

of brightness temperature with central time 12 UTC, and use the CMCC to track motion (i.e. find the grid offsets that best explain the change of imager intensity) from one day to the next. As a result, what we measure are the two components $\mathbf{d}X$ and $\mathbf{d}Y$ along the EASE2 projection of the time extensive (net Lagrangian) 24 h displacement starting at 12 UTC (on average). This is why our main variables are $\mathbf{d}X$ and $\mathbf{d}Y$, expressed with units km. For the validation, we compute buoy-equivalent displacement vectors between two position records of a buoy, and report validation statistics as bias and RMSE of $\mathbf{d}X$ and $\mathbf{d}Y$ in km. This allows us to assign uncertainties to the $\mathbf{d}X$ and $\mathbf{d}Y$ components of each vector, again with units km. The chosen representation is observation-driven, the closest to how we actually measure sea-ice motion. All other choices of components or units would have been through a transformation, and would have hidden away the nature of the observation system. Before addressing what this means for a user, we highlight one aspect of the production chain for which we were not fully consistent, namely the free-drift model. We indeed reproject and rotate ERA5 winds to the same EASE2 projection as the CDR thus we do operate the free-drift model along the same directions as the CMCC. However we tune the model using (Eulerian) averaged wind velocities (from 12 UTC to 12 UTC the next day) and tune it against (net Lagrangian) displacement vectors derived with the CMCC. When running the model, we enter the same daily (Eulerian) averaged wind velocities and thus obtain daily (Eulerian) averaged sea-ice velocities that we transform to displacements by multiplying the components by 24 hours. The more correct way would have been to integrate the hourly winds from ERA5 to yield daily sea-ice displacement vectors, both when tuning and running the free-drift model. We highlight this inconsistency for the sake of transparency and for the thought exercise. We do not consider this inconsistency to be a major source of error in the wind-derived sea-ice displacement vectors.

Users of the OSI SAF CDR might not find our $\mathbf{d}X$ and $\mathbf{d}Y$ components in km to be practical for their applications. Some might need the velocity vectors along a different projection. Others might prefer to access velocities in the meteorological convention, i.e. along the zonal and azimuthal components. Others might prefer to access sea-ice speeds and directions. All these representations are accessible via transformations of the $\mathbf{d}X$ and $\mathbf{d}Y$ components, keeping in mind the caveats mentioned above. We provide a Python3 notebook to illustrate some of these transforms (Lavergne, 2023), including computing weekly and monthly mean sea-ice motion fields.

## 4.3 Results from validation against drifter data

We present here the results of the validation of the sea-ice drift CDR, using the in-situ data introduced in Sect. 2.5 and the filtering and collocation strategies presented in Sect. 3.6. Results are first presented for individual singe-sensor satellite products (winter seasons only), then for the wind-driven drift product (summer seasons only), and finally for the merged multi-source product (year-round). These results are extracted from the dedicated Scientific Validation Report (SVR) (Lavergne and Down, 2022c).

### 4.3.1 Single-sensor satellite-based products (winter)

Figure 6 show validation scatterplots for three selected single-sensor satellite-based sea-ice motion fields: from SSM/I F11, SSMIS F17, and AMSR2 GW1. The panels illustrate the overall very good agreement between the satellite drift products and the displacement estimated from the buoy trajectories over the lifetime of the satellite missions. The validation data pairs are

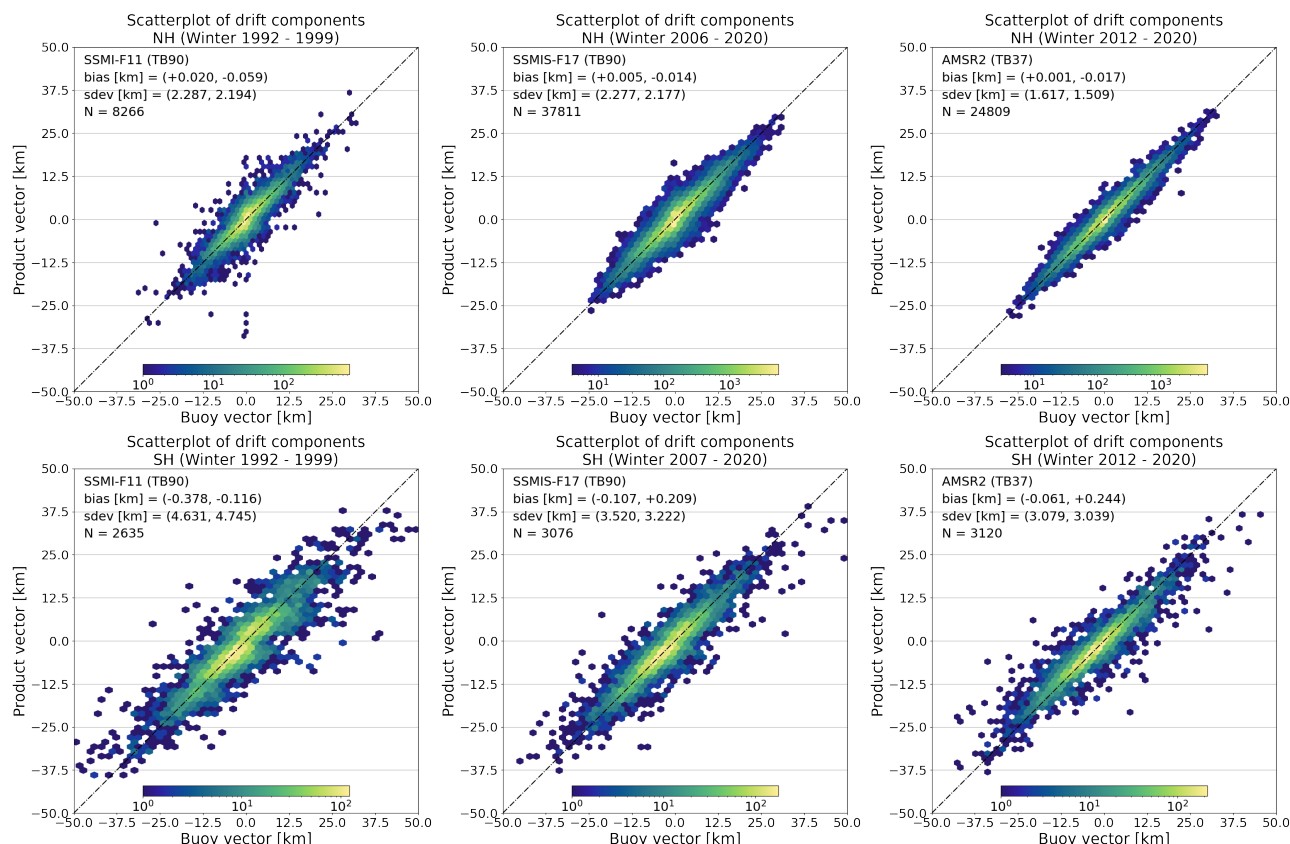

**Figure 6.** Hexagon-binned scatter plots of validation pairs for the full missions of SSM/I F11 (left), SSMIS F17 (middle), and AMSR2 GW1 (right) single-sensor products for winter seasons in the Northern (top) and Southern (bottom) Hemispheres. X-axis: components of the drift vectors from the buoys ($\mathbf{d}X$ and $\mathbf{d}Y$ together), Y-axis: components of the drift vector from the satellite ($\mathbf{d}X$ and $\mathbf{d}Y$ together). The bias and RMSE in $\mathbf{d}X$ and $\mathbf{d}Y$, as well as the total number of matchups $N$ are reported in the plot area. Note the changing logarithmic scale used for the colormap representing the density of samples in each cell. Only winter seasons.

concentrated and aligned with the 1-to-1 line. The three products return better validation statistics (lower bias and RMSE) in the Northern than in the Southern Hemisphere. AMSR2 GW1 (using the 36.5 GHz imagery) returns more accurate sea-ice drift vectors than SSM/I and SSMIS (using the near-90 GHz imagery). This is consistent with the validation results obtained with the near-real-time OSI SAF sea-ice drift product OSI-405, and with the results in Lavergne et al. (2010) (where AMSR-E Aqua was used instead of AMSR2 GW1).

Table 4 and 5 summarize the validation statistics for all the satellite-based single-sensor products entering the sea-ice drift CDR. The statistics are the same as those already reported in Fig. 6 and show better accuracy (lower RMSEs) for the AMSRs than for the SSM/I and SSMIS missions. One can also see a slight improvement from the early SSM/I to the later SSMIS, with Northern Hemisphere RMSEs going from about 4 km for the SSM/I to 3.5 km for the SSMIS. One should however be cautious

| Product | Season | $\langle \varepsilon(\mathbf{d}X) \rangle$ | $\langle \varepsilon(\mathbf{d}Y) \rangle$ | $\sigma(\mathbf{dX})$ | $\sigma(\mathbf{dY})$ | $N$ |
|---|---|---|---|---|---|---|
| ssmi-f10 | W | -0.019 | -0.050 | **2.718** | **2.678** | 5803 |
| ssmi-f11 | W | +0.020 | -0.059 | **2.287** | **2.194** | 8266 |
| ssmi-f13 | W | -0.032 | +0.002 | **2.528** | **2.660** | 19900 |
| ssmi-f14 | W | +0.043 | +0.009 | **3.017** | **2.605** | 14492 |
| ssmi-f15 | W | +0.004 | +0.003 | **2.284** | **2.263** | 8464 |
| ssmis-f16 | W | -0.009 | -0.023 | **2.572** | **2.385** | 42188 |
| ssmis-f17 | W | +0.005 | -0.014 | **2.277** | **2.177** | 37811 |
| ssmis-f18 | W | +0.007 | -0.026 | **2.206** | **2.007** | 28382 |
| amsr-aq | W | -0.001 | -0.019 | **2.024** | **1.999** | 21051 |
| amsr2-gw1 | W | +0.001 | -0.017 | **1.617** | **1.509** | 24809 |

**Table 4.** Winter validation statistics in the Northern Hemisphere for all the satellite-based single-sensor products entering the sea-ice drift CDR. The biases and RMSEs in $\mathbf{d}X$ and $\mathbf{d}Y$ are reported as well as the total number of buoy matchups $N$.

| Product | Season | $\langle \varepsilon(\mathbf{d}X) \rangle$ | $\langle \varepsilon(\mathbf{d}Y) \rangle$ | $\sigma(\mathbf{dX})$ | $\sigma(\mathbf{dY})$ | $N$ |
|---|---|---|---|---|---|---|
| ssmi-f10 | W | -0.217 | +0.034 | **4.122** | **4.101** | 1969 |
| ssmi-f11 | W | -0.378 | -0.116 | **4.631** | **4.745** | 2635 |
| ssmi-f13 | W | -0.274 | -0.050 | **4.084** | **4.502** | 3118 |
| ssmi-f14 | W | -0.304 | +0.004 | **4.406** | **4.594** | 2044 |
| ssmi-f15 | W | -0.317 | +0.328 | **3.646** | **3.471** | 809 |
| ssmis-f16 | W | -0.057 | +0.196 | **3.683** | **3.969** | 3056 |
| ssmis-f17 | W | -0.107 | +0.209 | **3.520** | **3.222** | 3076 |
| ssmis-f18 | W | -0.033 | +0.210 | **3.298** | **3.176** | 2978 |
| amsr-aq | W | -0.120 | +0.226 | **2.167** | **2.243** | 351 |
| amsr2-gw1 | W | -0.061 | +0.244 | **3.079** | **3.039** | 3120 |

**Table 5.** Same as Table 4 but for the Southern Hemisphere.

in attributing this improvement only to more recent satellite missions : one should also consider that the more recent part of the validation period has more buoys reporting with hourly GPS positions whereas the early period had only Argos-positioned buoys with 3-hourly trajectories.

Whereas the bias is nearly zero for all satellite missions in the Northern Hemisphere, it can take larger values (max 0.3 km) in the Southern Hemisphere for some of them. This larger bias can be observed in the bottom row of Fig. 6 where the highest

drift components (absolute value larger than 25 km) show an underestimation by the satellite product while the core of the validation sample is along the 1-to-1 line. The Southern Hemisphere bias thus seems related to highly dynamic drift conditions that are difficult to capture from the 10-15 km resolution satellite imagery used as input.

A comparison of Table 4 and Table 5 also confirms that there are much fewer buoy data (an order of magnitude less) available for our validation in the SH, and that we particularly lack SH buoy data for some of the satellite sensors including AMSR-E
Aqua and SSM/I F15 (see Fig. 1).

In any case, we note that both the RMSE (few kms) and the bias (a few hundred meters at max) are a fraction of the range covered by the natural variability of daily sea-ice drift (tens of kilometers).

### 4.3.2 Wind-driven product (summer)

Figure 7 shows the matchup of the wind vectors against the buoy data during the summer period for the Northern and Southern
Hemisphere, for the whole sea-ice drift CDR period (1991-2020), thus a longer period than it was tuned for (2002-2020, see Sect. 3.3). As was the case with the satellite-based products, the matchups are generally concentrated along the 1-to-1 line. However, the cloud of points are not well aligned with the diagonal of the plot and we find that the wind-based product is not as dynamic as the buoy motion. Still, the validation statistics for the wind-driven drift during summer (Table 6) are generally of the same order as those for the satellite-based drift in winter, with RMSEs around 3 km.

We note that the bias of the wind-driven fields take on slightly larger values than the satellite-based fields, but the average bias over the 30 years period is limited (this claim will be revisited in a later section). The wind-driven drift product thus seems to represent a useful complement to the satellite-based drift products as it extends the coverage into summer. This is particularly important for applications such as Lagrangian sea-ice tracking (and thus sea-ice age estimation).

| Hemisphere | Season | $\langle \varepsilon(\mathbf{d}X) \rangle$ | $\langle \varepsilon(\mathbf{d}Y) \rangle$ | $\sigma(\mathbf{dX})$ | $\sigma(\mathbf{dY})$ | $N$ |
|---|---|---|---|---|---|---|
| era5-wind (NH) | S | -0.145 | +0.085 | **2.578** | **2.500** | 41365 |
| era5-wind (SH) | S | +0.044 | +0.230 | **3.164** | **2.925** | 2930 |

**Table 6.** Summer validation statistics in the Northern and Southern Hemisphere for the wind-based product. The biases and RMSEs in $\mathbf{d}X$ and $\mathbf{d}Y$ are reported as well as the total number of buoy matchups $N$.

### 4.3.3 Multi-source merged product (year-round)

The multi-source product is the entry point for many users of the CDR. As described in Sect. 3.4, the single-sensor drift products and the wind-drift product are merged together using their respective uncertainties as weight. In addition, remaining gaps in coverage are filled with spatial interpolation to provide daily-complete products throughout the seasons and the 30 years period. The year-round validation histogram for the multi-source merged product is in Fig. 8, while Table 7 and Table 8 summarize the validation statistics for the whole 30 years for winter (W), summer (S), and year-round (Y).

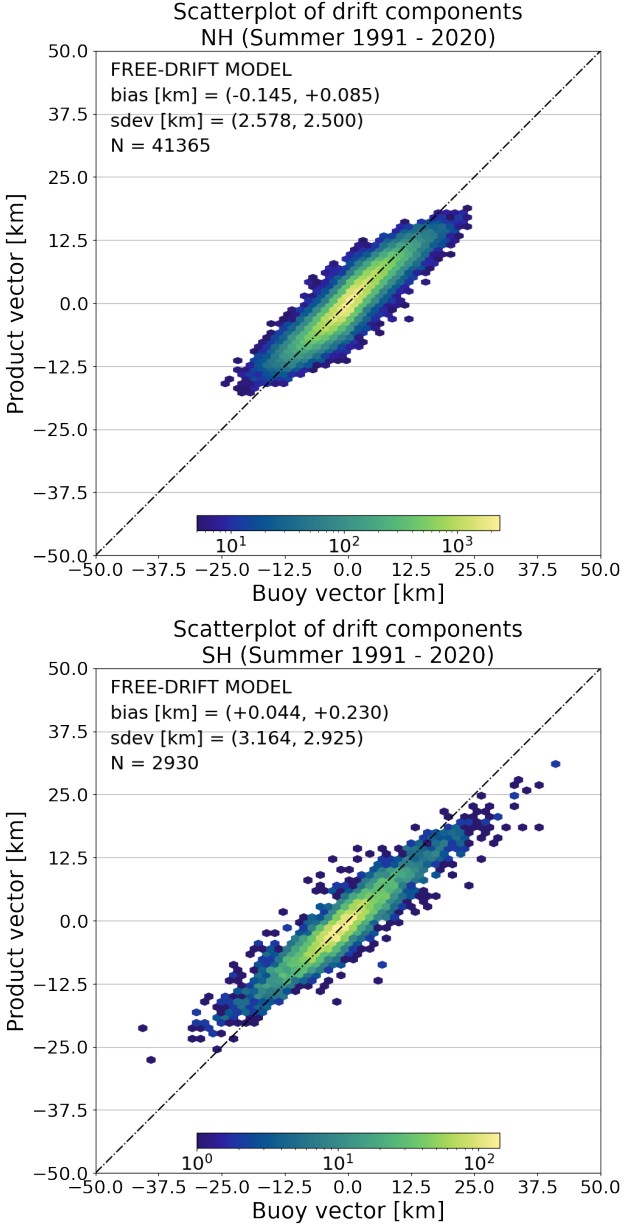

**Figure 7.** Hexagon-binned scatter plots of validation pairs for the wind-driven motion product for summer seasons in the Northern (top) and Southern (bottom) Hemispheres over the CDR period. X-axis: components of the drift vectors from the buoys ($\mathbf{d}X$ and $\mathbf{d}Y$ together), Y-axis: components of the drift vector from the free-drift model ($\mathbf{d}X$ and $\mathbf{d}Y$ together). The bias and RMSE in $\mathbf{d}X$ and $\mathbf{d}Y$, as well as the total number of matchups $N$ are reported in the plot area. Note the changing logarithmic scale used for the colormap representing the density of samples in each cell. Only summer seasons.

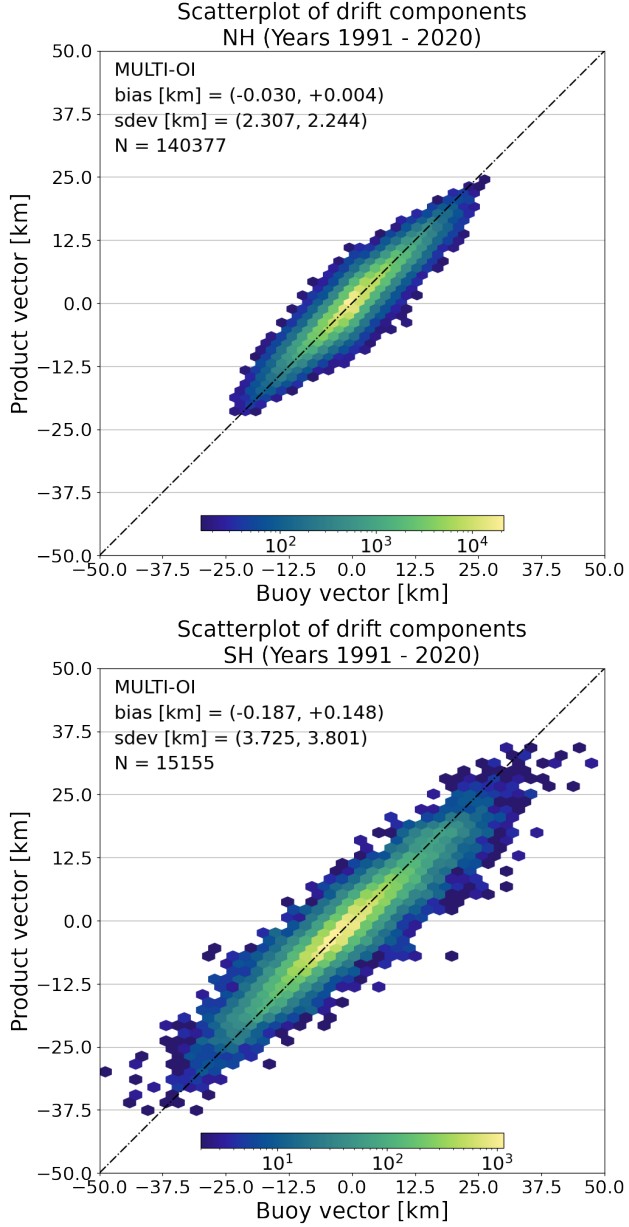

**Figure 8.** Hexagon-binned scatter plots of validation pairs for the multi-source merged product for Northern (top) and Southern (bottom) Hemispheres over the CDR period, year-round. X-axis: components of the drift vectors from the buoys ($\mathbf{d}X$ and $\mathbf{d}Y$ together), Y-axis: components of the drift vector from the merged product ($\mathbf{d}X$ and $\mathbf{d}Y$ together). The bias and RMSE in $\mathbf{d}X$ and $\mathbf{d}Y$, as well as the total number of matchups $N$ are reported in the plot area. Note the changing logarithmic scale used for the colormap representing the density of samples in each cell. Year-round statistics.

| Product | Season | $\langle\varepsilon(\mathbf{d}X)\rangle$ | $\langle\varepsilon(\mathbf{d}Y)\rangle$ | $\sigma(\mathbf{d}X)$ | $\sigma(\mathbf{d}Y)$ | $N$ |
|---------|--------|------|------|------|------|------|
| multi-oi | W | +0.014 | -0.024 | **2.108** | **2.031** | 70038 |
| multi-oi | S | -0.145 | +0.085 | **2.578** | **2.500** | 41365 |
| multi-oi | Y | -0.030 | +0.004 | **2.307** | **2.244** | 140377 |

**Table 7.** Validation statistics for the Northern Hemisphere multi-source merged product. The biases and RMSEs in $\mathbf{d}X$ and $\mathbf{d}Y$ are reported as well as the total number of buoy matchups $N$. "W" means winter, "S" means summer, and "Y" means all year (see Table 2).

| Product | Season | $\langle\varepsilon(\mathbf{d}X)\rangle$ | $\langle\varepsilon(\mathbf{d}Y)\rangle$ | $\sigma(\mathbf{d}X)$ | $\sigma(\mathbf{d}Y)$ | $N$ |
|---------|--------|------|------|------|------|------|
| multi-oi | W | -0.278 | +0.147 | **3.952** | **4.054** | 9572 |
| multi-oi | S | +0.032 | +0.255 | **3.183** | **2.985** | 2938 |
| multi-oi | Y | -0.187 | +0.148 | **3.725** | **3.801** | 15155 |

**Table 8.** Same as Table 7 for the Southern Hemisphere.

As expected, the multi-oi winter RMSEs are within the range of the single-sensor RMSEs, confirming that the optimal merging strategy implemented in the merging procedure is able to use the input single-sensor products to provide consolidated fields of winter sea-ice drift. In summer, the multi-oi RMSEs are equal to those obtained with the wind-driven drift product, since the latter is the only contribution to the multi-oi fields during summer.

The year-round RMSEs for the multi-oi CDR are of 2.5 km in the Northern Hemisphere and 3.9 km in the Southern Hemi-
sphere over the 30 years period. The year-round biases are near zero in the Northern Hemisphere and more pronounced (less than 200 m) in the Southern Hemisphere.

These values should also be interpreted in comparison to the original spatial resolution of the microwave imaging channels used to process the sea-ice drift vectors which is in the order of 10-15 km in diameter, a confirmation that the CMCC algorithms achieves sub-pixel motion accuracy, as found by Lavergne et al. (2010). Finally, we note that the RMSE of a coarse resolution
(75x75 km) product against point-like observations from buoys includes an (unknown) amount of representativity uncertainty, so that the actual RMSE of the CDR to equivalent resolution "truth" drift would be smaller.

### 4.3.4 Temporal evolution of the validation statistics over the length of the CDR (Arctic)

We now investigate the temporal evolution of the validation statistics (RMSE and bias) over the full length of the timeseries. We limit this analysis to the Arctic Ocean because the irregular buoy coverage in the Southern Hemisphere and its decade long
gap and regional heterogeneity (Fig. 1) severely limits any interpretation.

Figure 9 plots the temporal evolution over the period 1991-2020 of the $\mathbf{d}X$ RMSE to buoy drift (top), the $\mathbf{d}X$ bias to buoy drift (bottom). Plots for the $\mathbf{d}Y$ components of the motion vectors are similar and not shown here (see SVR, Lavergne and

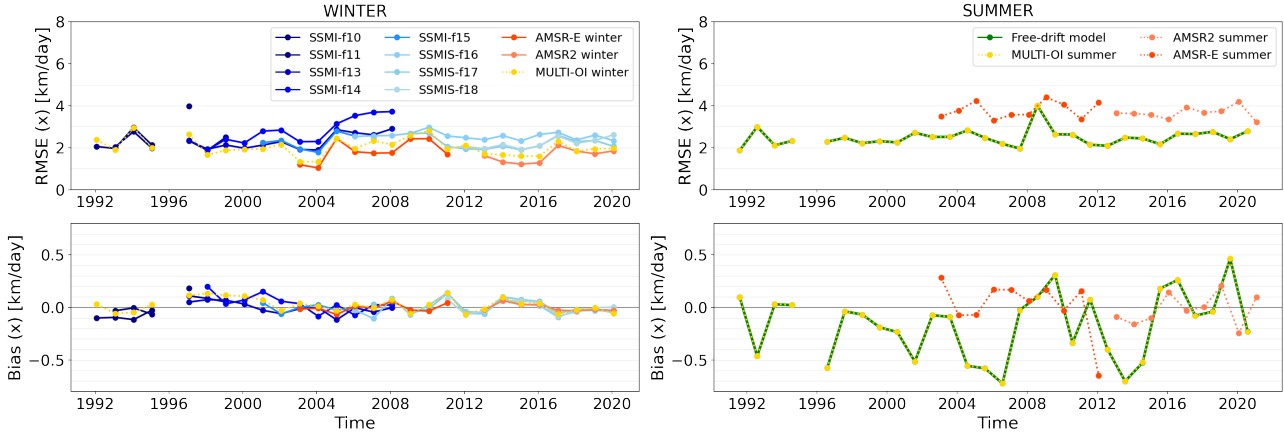

**Figure 9.** Timeseries of yearly validation statistics (top: RMSE, middle: bias) for the $\mathbf{d}X$ component of the drift vector in the winter (left) and summer (right) seasons in the Northern Hemisphere. It the top and middle panels, line colors correspond to the multi-sensor CDR (yellow) and the single-sensor products (other colours).

Down (2022c)). RMSEs and biases for the winter seasons (left) are stable across the lifetime of satellite missions. The SSM/I and SSMIS missions (blue lines) achieve winter RMSEs around 3 km, while the AMSR-E and AMSR2 missions achieve winter RMSEs lower than 2 km (red and orange lines). The multi-source merged product remains stable around 2 km winter RMSE (yellow line). These plots show that the winter bias stays constrained around 0 throughout the time period.

The right column of Fig. 9 tells a similar story for the summer RMSEs of the wind-driven motion, which stay stable around 2.5 km over the 30 years of the CDR. Noticeably, the bias (bottom panel) has much larger values than for the satellite-based drift fields. On individual years, the bias reaches -0.5 km in the first half of the timeseries, but gets closer to 0 (on average) in the last part of the timeseries. Similar large biases are seen for the $\mathbf{d}Y$ component of the drift vector (not shown). The larger bias during the summer season will be further discussed in Sect. 6.2. The right column of Fig. 9 also shows the RMSE and bias for the summer sea-ice motion fields obtained from the AMSR-E and AMSR2 missions. These satellite-based fields from the summer season do not directly enter the CDR, but are used as a target for tuning the free-drift model (Sect. 3.3.1). The AMSR-E and AMSR2 motion fields achieve a worse RMSE and bias than during winter (compare right and left panels). The summer wind-driven motion achieves a better RMSE than the satellite-based motion against which it is tuned, but the bias is significantly larger.

## 4.4 Data access and citation

The Digital Object Identifier (DOI) for this CDR is https://doi.org/10.15770/EUM_SAF_OSI_0012. Data files are also accessible from the OSI SAF High-Latitude FTP server (ftp://osisaf.met.no/reprocessed/ice/drift_lr/) and on MET Norway's THREDDS server (https://thredds.met.no/thredds/osisaf/osisaf_cdrseaicedrift.html), where the files are served with a number of protocols like HTTPS, OpenDAP, Subsetter, etc...

Data files are organized in `YYYY/MM/` catalogues where `YYYY` and `MM` are the year and month corresponding to the valid time of the product file (thus the end time of the displacement, see Sect. 4.1).

Product filenames follow the following convention:

`ice_drift_<area>_ease2-750_cdr-v1p0-<source>_24h-<date12>.nc`

`<area>` is either `nh` or `sh`. `<source>` describes the source of the motion vectors, e.g. `amsr2-gw1`, `ssmi-f13`, `era5-wind` (see the complete list in the left column of Table 1). `<source>` is the empty string for the muti-source merged product. `<date12>` is the valid date for the product file (thus the end date for the displacement) in format `YYYYMMDD1200`.

The CDR is also known as OSI-455 (OSI SAF product identifier) and is described at https://osi-saf.eumetsat.int/products/
osi-455, from where the user documents are available: Algorithm Theoretical Basis Document (Lavergne and Down, 2022a), Product User's Manual (Lavergne and Down, 2022b), and Scientific Validation Report (Lavergne and Down, 2022c).

We request that usage of this CDR is acknowledged by citing the data itself and the present paper. A BibTeX entry for the CDR can be retrieved from https://www.doi2bib.org/.

As for all its products, the OSI SAF team welcomes and values questions and feedback from users. Contact the authors or
visit https://osi-saf.eumetsat.int.

## 5   Comparison to existing CDRs

We conduct here a comparison of the OSI SAF CDR to other similar CDRs. The comparison is mostly in terms of product characteristics (temporal coverage, spatial resolution, etc...). We also compare validation results reported by other investigators to ours.

### 5.1   Temporal coverage

This first version of the OSI SAF sea-ice drift CDR covers the period January 1991 through December 2020 (year-round) and starts with the SSM/I F10 mission (Table 1).

To extend the CDR to cover earlier satellite missions is not straightforward. The SSM/I F08 mission (June 1987 through December 1991) carried 85.5 GHz imagery channels similar to those of F10 and later missions. However, these channels
degraded rapidly and are regarded as defective on F08 from April 1988 onward (Hollinger et al., 1990; Fennig et al., 2020). SMMR (October 1978 through August 1987) did not have similar high-frequency channels and only offered imagery with 30 to 50 km spatial resolution at best with the 37 GHz imagery. To extract sea-ice motion from such coarse resolution imagery is challenging as it may result in larger uncertainties and biases. Another challenge is that the SMMR period only brings sea-ice imagery every other day (except for the first 20 days of the mission in Oct./Nov. 1978), and with a large polar observation hole
(north of 84° N).

Tschudi et al. (2020) present the NSIDC sea-ice motion dataset v4 that starts in 1978. However, they note how much more uncertain the first decade is for the reasons stated above. Haumann et al. (2016) reports on extensive quality-control and data filtering and correction to remove artificial jumps and trends in the first decade of the NSIDC data record (see their section

*Time-series homogenization* and particularly their Extended Data Figure 4). Sun and Eisenman (2021) use the NSIDC data record from 1992 onward because of biases they find in the earlier period (their Supplementary Figure 3).

Kwok et al. (1998) also processed vectors before 1991, but they did not attempt to merge them into a full-length CDR. The dataset of Girard-Ardhuin and Ezraty (2012) starts in January 1991 for the Arctic, and 2003 for the Antarctic (from AMSR-E and AMSR2 missions only).

One of the positive characteristics of the OSI SAF sea-ice drift CDR presented here is thus that the same sources of sea-ice motion are used in the Arctic and the Antarctic, and that the type of satellite imagery is stable through the three decades covered.

The NSIDC data record is a year-round dataset, using a free-drift model driven by National Centers for Environmental Prediction (NCEP) wind data. Both the Kwok et al. (1998) and IFREMER datasets are winter-only.

## 5.2 Grid spacing and spatial resolution

The product grid has a 75 x 75 km spacing. This is selected to be exactly six times the grid spacing on which we prepare the satellite imagery (12.5 km), thus we compute drift vectors every $6^{th}$ image pixels along both x- and y-axis. However, the sub-images (*aka* blocks) used in the motion tracking step have a diameter of about 100 km (Sect. 3.2). As a result, neighbouring vectors make use of partly overlapping image blocks, and are thus not independent from each others. In other words, the grid spacing of 75 km leads to a moderate oversampling and the true spatial resolution of the motion field is somewhat coarser, closer to 100 km. We also selected EASE2 grids with a 75 km spacing to align with the EASE2 25 km grids used by Lavergne et al. (2019) for their SIC CDRs (a sea-ice drift vector every 3x3 SIC grid cells).

Our grid spacing is thus significantly coarser than that of the NSIDC data record (25 km, Tschudi et al. (2020)), somewhat coarser than that of the IFREMER timeseries (62.5 km and 31.25 km) (Girard-Ardhuin and Ezraty, 2012), and finer than that selected by Kwok et al. (1998) (100 km). However, since passive microwave imagery from SSM/I and SSMIS with a resolution of 12.5 km at best is the core source of all these data records, it is not clear if their finer grids translates into better spatial resolution (in terms of information content). For example, the NSIDC data record only offers the 25 km grid spacing in the multi-sensor merged fields, but all the input motion fields are at a coarser resolution, in particular the SSM/I and SSMIS fields are at 75 km grid spacing, like ours (Tschudi et al., 2020, Table 2).

## 5.3 Drift duration

The CDR holds daily sea-ice drift vectors. The duration of each drift vector is roughly 24h from 12 UTC to 12 UTC, but the actual start and end times (and thus durations) of each vector is provided in the product file, since they vary across the product grid and from one day to the next (see Fig. 4).

The NSIDC data record gives also access to daily vectors, but these might suffer from remaining quantization noise. This is the stated reason why they recommend their weekly averaged product for most applications (Tschudi et al., 2020, bottom of section 2.3).

The IFREMER data record has different temporal resolutions depending on the input satellite data source, and hemisphere. Their main Arctic 30-year CDR from passive microwave and scatterometer missions holds 3-day vectors. This is because they use the MCC algorithm from the nominal 12.5 km imagery. A shorter drift duration would result in more quantization noise. Their shorter Antarctic dataset is based on AMSR-E and AMSR2 missions and is available both as 2-day and 3-day vectors.

Table 9 summarizes the main characteristics of the OSI SAF, NSIDC, and IFREMER sea-ice drift CDRs.

| | Input data sources | Period covered | Year-round | Extension | Grid & Spacing | Drift Duration |
|---|---|---|---|---|---|---|
| OSI SAF CDR v1 | PMW, NWP winds | 1991-2020 | Yes | No | EASE2 75 km | daily |
| NSIDC CDR v4 | PMW, NWP winds, Opt/IR (NH 1981–2000), Buoys (NH only) | 1978 -> | Yes | Yearly | EASE 25 km | daily weekly |
| IFREMER | PMW, SCATT (NH only) | NH: 1991 -> SH: 2003 -> | Winter | Monthly | Pol. Stereo. 62.5 km (31.25 km in SH) | 2-, 3-, 6-, and 30-days |

**Table 9.** Summary table of the main characteristics of the OSI SAF, NSIDC, and IFREMER sea-ice drift CDRs at time of writing. PMW stands for Passive Microwave, NWP for Numerical Weather Prediction, Opt/IR for Optical and Infrared, SCATT for Scatterometry, NH for Northern Hemisphere, SH for Southern Hemisphere.

## 5.4 Accuracy against buoys

Assessing and comparing the accuracy of the existing sea-ice drift CDRs would require a dedicated intercomparison exercise which is beyond the scope of this manuscript. As for all intercomparisons, the task is not straightforward as the various CDRs have different characteristics (see Table 9) that require careful consideration. Because the geographical projections are different, the lengths and directions of the components of the drift vectors cannot necessarily be compared. Also, the drift duration differs and one must decide how to compare, e.g. a daily drift vectors with a 3-day vector. Finally, the choice of the validation metrics and the units to report them is critical 4.2. These reasons and others contribute to why sea-ice drift intercomparison is tackled differently by different authors (see e.g. Hwang (2013); Sumata et al. (2014); Wang et al. (2022)), and conclusions generally differ.

With the limitations above in mind, we compare here our validation results with those published by other authors. We underline that this is only to provide a general idea of the accuracy of the new CDR relative to other datasets and not a replacement for a (future) proper intercomparison. In Table 7, we report an RMSE of 2.3 km for the Northern Hemisphere for the components of the daily drift vector for the full length of the CDR.

Tschudi et al. (2020) report an RMSE of about 4 cm.s$^{-1}$ when comparing their v4 merged daily motion vectors against buoys from the CRREL Ice Mass Balance Buoy program in the Northern Hemisphere over the period 2000-2016. Considering a nominal drift duration of 24 h, this translates into 3.5 km. They additionally report RMSEs of the daily single-sensor products

in year 2005, e.g. 3.2 km (3.7 cm.s$^{-1}$) for winter AMSR-E data, 4.1 km (4.7 cm.s$^{-1}$) for winter SSM/I 85 GHz data, and 3.7 km (4.3 cm.s$^{-1}$) for summer wind-derived data. For comparison, we obtain 2.0 km (AMSR-E, winter), 2.0-3.0 km (various SSM/I missions, winter), and 2.5 km (Winds, summer). All in all, our validation statistics are substantially lower than those reported
for the NSIDC CDR for comparable missions and for the merged daily product. Tschudi et al. (2020) do not report RMSEs for the Southern Hemisphere due to the limited number of buoys. Girard-Ardhuin and Ezraty (2012) do not report validation statistics in terms of x and y components of the displacement vectors, but only in terms of magnitude and direction. Their values are thus not comparable to ours.

## 6    Discussion, known limitations and outlook

The sea-ice drift CDR presented here is the first version of such CDR in the context of the EUMETSAT OSI SAF. Although we trust the 30-years global timeseries is a useful contribution to climate science, we also warn its users about known limitations and the possible impacts of the validation results presented in Sect. 4.3. We then outline possible R&D activities to improve future versions of our CDR.

### 6.1    Negative bias for highly dynamic motion (Southern Hemisphere)

Table 5 documents higher overall low bias of the satellite-based drift vectors in the Southern Hemisphere than in the Northern Hemisphere (Table 4). Figure 6 confirms it mostly originates from the longer drift components, those with absolute values of buoy drift larger than 25 km in 24 hours, thus larger than 30 cm.s$^{-1}$ over 24 hours. For these highly dynamic motion, the satellite-based component is biased low. This probably originates from enforcing a maximum drift speed in the motion tracking step (Sect. 3.2). Although we chose this maximum drift speed to 0.45 cm.s$^{-1}$ (sustained over 24 hours), this threshold might
still impact the CMCC optimization and nudge it towards smaller drift components. Still, we recall that each vector in the CDR represents the mean motion of a large ( 100 x 100 km$^2$) area of the sea-ice cover while the buoy drift is a point-like observation (the individual floe). The difference of spatial scales might also explain some of the mismatch at high speed. Although we only see a significant amount of such high speed drift components in the Southern Hemisphere, they can also be encountered in the Northern Hemisphere, although less frequently.

Contrarily to with the MCC, to increase the maximum allowed drift speed does not necessarily result in a longer processing time of the CMCC. In future versions of the CDR, we will investigate if the validation of these highly dynamic sea-ice motion improves when the maximum allowed speed is set to a higher value, possibly combined with the two-step motion tracking approach introduced in Sect. 5.2.

### 6.2    Negative bias for wind-derived motion (summer season)

The original plan for this first version of the OSI SAF sea-ice drift CDR was to prepare a winter-only data record. It however rapidly became evident that some key applications would not be possible, e.g. tracking long trajectories (Krumpen et al., 2019;

Sumata et al., 2022) or mapping sea-ice age (Tschudi et al., 2020; Korosov et al., 2018). A free-drift model approach was thus investigated and adopted for bridging the summer season gaps in satellite retrievals.

Figure 9 (right column) however documents the negative summer bias in the $\mathbf{d}X$ component (same for $\mathbf{d}Y$, not shown). We have no definite explanation for this bias, nor for why it reduces in the later part of the CDR (2016-2020). We recall that the free-drift model was tuned against satellite-based sea-ice motion from the AMSR-E and AMSR2 missions, thus in the period 2002-2020 (Sect. 3.3). Firstly, validation of the summer sea-ice motion data from AMSR-E and AMSR2, not included in the CDR but shown on Fig. 9, exhibits a larger bias than in the winter season. This bias in satellite-derived motion is amplified into the free-drift model parameters and the predictions from that model. Second, it is well documented that sea-ice motion has experienced a positive trend over the last decades, concomitant with sea-ice thinning in the Arctic (Rampal et al., 2009; Spreen et al., 2011; Kwok et al., 2013; Olason and Notz, 2014) and stronger winds in the Antarctic (Holland and Kwok, 2012; Kwok et al., 2017). Since we tune our free-drift model against data from the later part of the period, where sea-ice motion is transitioning to a faster regime, we might have difficulties to capture these trends in our summer drift. We do not have access to a good enough buoy coverage for the Southern Hemisphere to assess the temporal evolution of the negative summer bias there, but an overall bias is present (Table 6). We finally note that we cannot rule-out that other factors, e.g. (hypothetical) trends in polar ERA5 winds, contribute to the observed trend in the bias of our summer drift.

Recently, Brunette et al. (2022) revisited the formulation and tuning of the free-drift model for Arctic sea-ice motion. One of their motivation was to improve summer sea-ice motion as provided in the NSIDC data record, but their study includes year-round sea-ice motion. The main novelty of their approach is the state-dependent free-drift model: they parametrize the wind-ice-ocean transfer coefficient on sea-ice thickness. They obtain their sea-ice thickness timeseries from a reanalysis of the coupled ocean-ice model system PIOMAS (Schweiger et al., 2011). They tune their free-drift model against buoy data directly, and not on a monthly basis. The spatial, seasonal and multi-decadal evolution of the PIOMAS sea-ice thicknesses translates into the spatial, seaonal and multi-decadal evolution of the wind-ice-ocean transfer coefficient $|A|$, and thus the sea-ice motion fields. They can also document significant changes in the retrieved under-ice current fields $U_{wg}$ before and after year 2000, especially in the southern branch of the Beaufort Gyre (their Fig. 9).

Our approach is more similar to that of Thomas (1999): we do not introduce a dependency on auxiliary sea-ice thicknesses, and rather rely on space-time-dependent tuning of the free-drift model parameters to capture the space-time-dependent relation between wind and sea-ice motion vectors. While this approach can capture the spatial and seasonal variability of such relation (Thomas, 1999), it cannot capture its multi-decadal evolution. Being tuned on the second part of the CDR period, our free-drift model cannot reproduce the ramp-up in summer sea-ice velocities from the first part of the period, nor the shift in under-ice currents around year 2000.

In conclusion, we bring to the attention of the users that the summer sea-ice drift fields might be biased over the 30-years period. In this first version, the summer wind-derived drift is more seen as gap-filler enabling some specific applications, rather than a basis for trend analysis. We do believe it is better for the user to access a year-round CDR with possibly some bias during the summer period, than a winter-only CDR. In any case, our summer sea-ice motion should be less biased than that in

the NSIDC timeseries that use a constant wind-ice-ocean transfer coefficient of $|A| = 0.01\%$ (compare to the values in Table 3), which captures neither the spatial, seasonal, nor multi-decadal evolution, as shown by Brunette et al. (2022).

## 6.3 Accuracy of sea-ice motion in coastal and peripheral seas

Partly by choice and partly because of the scarcity of in situ data, the validation statistics presented in Sect. 4.3 and discussed earlier exclude coastal sea-ice motion as well as peripheral sea-ice area such as Hudson Bay, Baffin Bay, Bering Sea and the Greenland Sea (see Fig. 1). Coastal sea-ice vectors are excluded from the validation by the collocation strategy, specifically to avoid that buoys deployed or having drifted in land-fast sea-ice region are compared to off-shore motion vectors that represent the motion of the pack ice. Of the peripheral sea-ice regions, the case of Greenland Sea merits a specific mention. Due to the general motion pattern in the Arctic Ocean, a fair number of in situ drifters eventually exit through Fram Strait and enter the Greenland Sea, where sea-ice motion can be very dynamic (because of smaller floes) and with strong spatial gradients (from fast-ice along Greenland towards the open ocean). Our experience with the OSI SAF near-real-time product is that such coarser resolution products based on existing passive microwave imagery does not validate well against buoy data because of this strong spatial gradient.

## 6.4 Outlook for the OSI SAF CDR

The back extension of the CDR to the start of the SMMR mission (Oct. 1978) will be desired feature for later releases. The crux will be to ensure no biases are introduced at the transition from using the coarse resolution SMMR and SSM/I 37 GHz imagery prior to 1991, to using the higher-resolution SSM/I 85.5 GHz imagery onwards. A routine forward-extension of the OSI SAF CDR (an Interim CDR) is also under study. In the meantime, users needing routine extension of the CDR might consider using the OSI SAF near-real-time product (OSI-405) although it does not come on the same grid (polar stereographic vs EASE2), nor with the same spacing (62.5 km vs 75 km), not time-span (48 hours vs 24 hours). Both use the CMCC algorithm and follow similar file formats.

We will also investigate if higher spatial resolution can be achieved from the given input satellite imagery. In sea-ice motion tracking algorithms, the diameter and spacing of the sub-image are parameters selected by the investigator. Smaller and denser sub-images will allow tracking more local motion, but can lead to higher uncertainties and more rogue vectors because fewer intensity patterns are tracked. A possible way forward for upcoming versions of this CDR would be to adopt a two (or more) stages motion tracking approach: the first stage would be similar to the one we adopted for the CDR, while the second would use smaller and denser sub-images and use the vector field from the first stage as an a-priori information (first-guess). This would allow a finer grid spacing and spatial resolution. A possible way to measure the true spatial resolution of the CDRs, and if it improves with the two-step approach, would be to conduct a spectral scale analysis.

In addition to improving on the product characteristics above, we might want to revisit the tuning strategy for the free-drift model. A new strategy could be to use buoy motion - combined with satellite-based motion - as a target for the tuning. This should bring the wind-derived drift closer to the buoy drift in the summer validation. It is not yet clear how to capture the multi-decadal evolution of the tuning parameters. The novel strategy of Brunette et al. (2022) is effective, but it also brings

an extra dependency on sea-ice thickness data that might not be as easily applicable in the Southern Hemisphere. Another
approach would be to use the vectors derived from the AMSR-E and AMSR2 missions as input to the merged product during
summer, instead of using them only to tune the free-drift model. This would however require careful consideration of the
temporal consistency of the CDR when satellite-based summer vectors are introduced at the start of the 2000s.

## 7 Conclusions

We introduce OSI-455, the first release of a sea-ice drift climate data record by the EUMETSAT Ocean and Sea Ice Satellite
Application Facility (OSI SAF): https://doi.org/10.15770/EUM_SAF_OSI_0012. The CDR covers both Arctic and Antarctic
sea ice, year round, and for the period 1991-2020. It pertains of daily product files holding maps of daily sea-ice motion vectors
with uncertainties and flags. The CDR uses an EASE2 projection with 75 km grid spacing.

The sea-ice drift CDR uses the Continuous Maximum Cross-Correlation (CMCC) method of Lavergne et al. (2010) on
daily-gridded maps of brightness temperature from the SSM/I, SSMIS, AMSR-E, and AMSR2 missions during the winter
months. A free-drift model is tuned using winds from the C3S/ECMWF ERA5 reanalysis to obtain sea-ice drift vectors during
the summer months. A multi-source merging algorithm optimally combines the different sources into the multi-source CDR
files.

The dataset is validated against an extensive collection of on-ice buoy trajectories. The validation results show that the winter
sea-ice drift RMSEs and biases are small compared to the range of variability of daily sea-ice drift. The winter RMSE is mostly
stable throughout the 30-years period of the CDR, and exhibits an improvement with more recent satellite missions. The winter
RMSE and bias are larger for the Antarctic than for the Arctic sea-ice motion. Both Arctic and Antarctic RMSEs and biases
are larger in the summer than in the winter season, and the bias is larger in the first half of the CDR. This is possibly because
the free-drift model was tuned against satellite sea-ice drift fields from the second half of the CDR. The summer sea-ice drift
vectors are thus appropriate for some applications (e.g. building long trajectories or estimating sea-ice age), but users should
be cautious with trend analysis of the summer sea-ice motion from this dataset.

This is the first release of a global sea-ice drift CDR by the EUMETSAT OSI SAF. We outline some areas of research that
could be investigated towards future versions.

## 8 Code and data availability

The resulting sea-ice drift climate data record is readily available as https://doi.org/10.15770/EUM_SAF_OSI_0012 (OSI SAF,
2022). Data files are also accessible from the OSI SAF High-Latitude FTP server (ftp://osisaf.met.no/reprocessed/ice/drift_lr/)
and from MET Norway's THREDDS server (https://thredds.met.no/thredds/osisaf/osisaf_cdrseaicedrift.html).

We request that usage of this CDR is acknowledged by citing the CDR and the present paper. A BibTeX entry for the CDR
can be retrieved from https://www.doi2bib.org/. In case no citation format is prescribed, we invite you to cite the dataset as:

OSI SAF Global Low Resolution Sea Ice Drift data record 1991-2020 (v1, 2022), OSI-455, doi: 10.15770/EUM_SAF_OSI_0012.
EUMETSAT Ocean and Sea Ice Satellite Application Facility. Data (for [extracted period], [extracted domain],) extracted on
[download date]


The monthly parameter files used to run the free-drift model (see Sect. 3.3) are made available in the same data catalogues
as the CDR itself. Software to reproduce the figures in this manuscript are made available as Down and Lavergne (2023a).
A notebook to demonstrate transformation and rotation of sea-ice drift vectors is provided (Lavergne, 2023). The processing

software for the CDR is available as Down and Lavergne (2023b).

## Appendix A:  Winter dates when the wind-driven motion field is used in the CDR

| Northern Hemisphere | Southern Hemisphere |
|---|---|
| 1991-01-01 – 1991-01-17 | 1991-03-02 – 1991-03-05 |
| 1991-01-22 – 1991-01-23 | 1991-03-09 – 1991-03-11 |
| 1991-01-28 – 1991-02-12 | 1991-03-19 – 1991-03-20 |
| 1991-02-16 – 1991-02-18 | 1991-03-26 – 1991-03-31 |
| 1991-02-24 – 1991-02-25 | 1991-04-01 – 1991-04-20 |
| 1991-03-03 – 1991-03-05 | 1991-04-30 – 1991-05-01 |
| 1991-03-10 – 1991-03-11 | 1991-05-06 – 1991-05-09 |
| 1991-03-27 – 1991-04-20 | 1991-05-20 – 1991-05-23 |
| 1991-04-30 | 1991-05-31 – 1991-06-01 |
| 1991-05-01 | 1991-06-11 |
| 1991-05-08 – 1991-05-09 | 1991-07-02 – 1991-07-03 |
| 1991-05-20 – 1991-05-21 | 1991-07-14 – 1991-07-16 |
| 1991-05-31 | 1991-07-25 – 1991-08-01 |
| 1991-12-06 – 1991-12-18 | 1991-08-07 – 1991-08-08 |
| 1993-01-04 | 1991-08-13 – 1991-08-19 |
| 1996-05-30 – 1996-05-31 | 1991-09-17 – 1991-09-20 |
| 2000-12-01 – 2000-12-02 | 1992-06-06 – 1992-06-08 |
|  | 1996-05-30 – 1996-05-31 |
|  | 1996-06-06 |

**Table A1.** Full days of the winter and transition seasons in the CDR which are gap-filled with wind-derived drift vectors due to lack of
passive microwave data.

*Author contributions.* TL designed the CDR and wrote most of the manuscript. ED developed and run the processing chain, run the validation software, and contributed to the analysis in the manuscript.

*Competing interests.* The authors declare no competing interests.

*Acknowledgements.* This research and the production of the CDR was funded through the EUMETSAT Ocean and Sea Ice Satellite Application Facility, specifically the third Continuous Development and Operations Phase (CDOP3, 2017-2022).

The Fundamental Climate Data Record (FCDR) Release 4 (10.5676/EUM_SAF_CM/FCDR_MWI/V004) from the EUMETSAT Climate Monitoring Satellite Application Facility (CM SAF) has been our source for SSM/I and SSMIS data (Fennig et al., 2020). We used the AMSR-E FCDR V003 (10.5067/AMSR-E/AE_L2A.003) of Ashcroft and Wentz. (2013), maintained by the National Snow and Ice Data 880 Center (NSIDC). We accessed the GCOM-W1 AMSR2 Level-1B files from the Japan Aerospace Exploration Agency (JAXA) G-portal. We accessed the 10-m wind fields from the C3S/ECMWF ERA5 reanalysis (Hersbach et al., 2020) on the C3S Climate Data Store.

We used a number of buoy trajectories for the validation of the CDR and acknowledge the programmes, institutions, and individuals that build and deploy them on sea ice. We also thank the programmes, institutions, and individuals that maintain online data archives for these buoys.

We acknowledge the contribution of Javier Mozo who investigated the free-drift model as part of a Master's thesis at the University of Oslo, and shared early versions of the software. We are thankful to our colleagues Cécile Hernandez (Météo France) and Steinar Eastwood (MET Norway) for easing the preparation and release of the CDR in the context of the OSI SAF, and to Atle Søresen (MET Norway) for accessing and pre-processing the satellite and ERA5 input data. We finally express our gratitude to Julien Nicolas and Hao Zuo (European Center for Medium-Range Weather Forecast), Gilles Garric (Mercator Ocean International), and Thomas Krumpen (Alfred Wegener Institute) who 890 provided valuable comments during the review cycle of the CDR.

The production and presentation of this CDR would not have been possible without Free/Libre and Open Source Software (FLOSS) including Python3 (Van Rossum and Drake, 2009), Numpy (Harris et al., 2020), Matplotlib (Hunter, 2007), Cartopy (Met Office, 2010 - 2015), pyresample (Hoese et al., 2022), and xarray (Hoyer et al., 2023), to mention a few. Background map data on Fig. 1, Fig. 2, Fig. 3, Fig. 4 and Fig. 5 are from Natural Earth Data (naturalearthdata.com).

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
