# Peer review of "A Climate Data Record of Year-Round Global Sea Ice Drift from the EUMETSAT OSI SAF"

_Earth System Science Data, 2023_

## Author Response (AR1)

Prepared by Thomas Lavergne and Emily Down (v1, Sept 5th 2023).

**Introduction to the point-by-point answer**

This document holds a point-by-point answer to all comments received during the initial review of ESSD-2023-40. We use colors to code our answers:

- June 3rd 2023 : Initial comment from 2 reviewers (RC1, RC2) and editor (EC1): Black
- June 28th 2023 : Our initial answers (AC): Blue
- July 8th 2023 : Editor's new comment (EC2): Red
- Sept 5th 2023 : Our point-by-point answers to all comments and edits (if any): Purple

The most prominent changes to the revised manuscript are:

In answer to Reviewer 1 and the Editor:

- We organized the existing text comparing our CDR to others like NSIDC and IFREMER into a dedicated section (Sect 5 : Comparison to other CDRs).
- We added a Table to summarize the noted differences (Table 9).
- We added a new subsection (5.4 Accuracy against buoys) in which we compare the validation statistics we obtain against buoys, to what other authors have reported on other CDRs earlier. This is not a new intercomparison done by us, but hopefully gives an idea of how our data validates relative to other CDRs.
- We placed the text providing an R&D outlook for the OSI SAF data record in a new section (Sect 6.4 Outlook for the OSI SAF CDR).

In answer to Reviewer 2 and the Editor:

- We authored a new section (Sect. 4.2 Vectorial representation of sea\=/ice drift) to justify our choice of focusing on the dX and dY components in the paper and in the product files. This text also touches on the difference between velocity and displacement.
- Acknowledging that some users will want to access velocities and/or other vector components, we prepared a notebook demonstrating how these transformations can be done from the OSI SAF CDR (https://zenodo.org/record/8315156).

**Reviewer comments RC1**

The authors introduced a new climate data record of sea ice drift vectors from 1991-2020, which covers both Arctic and Antarctic. This provides good continuity data for long time series observation of sea ice motion in polar regions, and is useful for quantifying the response of sea ice motion to climate change. The overall organization of this manuscript is clear and the methods are well described in detail. However, the comparative analysis of the product with other products in the manuscript is relatively vague and there is a lack of elaboration on the areas in which the product could be applied. After refining the above

deficiencies, I consider this article would be more in line with the main purpose of the *Earth System Science Data*. Some improvements could be made, as listed below.

Thank you for reviewing our manuscript and for your comments. We answer below.

Major comments:

1.    The introduction describes relatively little about the importance of sea ice drift, although much of it is well known. For readers unfamiliar with the field, what is the need to produce sea ice drift products? What can we apply sea ice drift products for? Please emphasize this in the introduction.

This is a good point. We might have taken this for granted. We will add more context in the introduction.

We added the following paragraph to the introduction.

*Sea ice in the polar regions can move several tens of kilometers per day under the actions of winds, ocean currents, and internal stresses. This motion redistributes sea ice from regions where it forms during winter, to regions where it melts during summer. Such redistribution of mass and fresh water has a profound influence on the polar oceans and their currents. Patterns of sea\=/ice motion define key characteristics of the sea\=/ice cover, such as the accumulation and re-circulation of thicker and older ice north for Canada and in the Beaufort Sea, or ice formation in latent heat polynyas. The distribution, export, and replenishment of sea\=/ice age is fully controlled by where sea\=/ice drifts. At a smaller scale, sea\=/ice motion steers the thickness distribution of sea\=/ice through the opening of leads and the formation of ridges.*

2.    Usually buoy data are considered to have a certain level of authenticity and accuracy, and the reason for not choosing buoy data as one of the input sources in the section on multi-source merging (Section 3.4) needs to be elaborated more than just emphasizing the difference from existing sea ice motion products (L285).

We agree this requires elaboration. Our choice was deliberated but is not justified. We will add a paragraph to cover this discussion in section 4.1 and we might remove the sentence L285.

We deleted the sentence L285 and added the following paragraph in section 4.1:

*Although buoys are a trusted source of sea\=/ice motion information and can capture high\=/resolution and high\=/frequency patterns that are not accessible from satellite missions we use, we do not include them as input to the merged multi\=/source product. This is conversely to what is done for the NSIDC sea\=/ice motion dataset, where buoys are used in the Arctic \citep{tschudi:2020:nsidcv4}. Our main motivation is to keep buoys as an independent source of validation data. Second, buoys are point-like observations while our satellite\=/based vectors represent the average motion of large areas. Combining the two scales in the merging was not straightforward. Although it improves in version 4 of the NSIDC data record, the inclusion of buoys has been shown to lead to non-physical discontinuities in the motion fields of earlier versions*

*\citep{szanyi:2016:nsidc_artefact,tschudi:2020:nsidcv4}.*

3.      The lack of comparisons with existing sea ice motion products would be relatively strange. For users of the data, a general overview comparison with widely used products is necessary. It is also useful to have a visual overview of the advantages of this new CDR product.

See our combined reply to your next point.

4.      After reading the whole manuscript, my main concern is, what exactly are the advantages of the new sea ice drift CDR product? This needs to be highlighted and detailed, but there is no summary paragraph in the introduction or in section 4.3 (Discussion, known limitations and outlook), which is not convincing. According to the manuscript, the new CDR has a coarse spatial resolution of 75 km and a time length of 1991-2020, which is not as fine as the products provided by NSIDC and IFREMER in terms of time length and spatial resolution.

We cover several of your points (spatial resolution, time length, spatial extent, and more) in our section 4.3, most of the time in comparison to NSIDC or IFREMER. Your comment however made us realize that we do not cover the temporal frequency aspects (daily for the new data, 3-days for IFREMER, 1 week for NSIDC) and we will add a subsection in 4.3.

We agree that at present we do not summarize this comparison, and that this could be useful for a user. We will thus add a summary table at the end of section 4.3. To improve readability we will also gather our text describing what could be improved towards the next version of our data in a new section, so as to focus 4.3 on the comparison to other data sources.

In this data-description manuscript, we report on an extensive validation against buoy trajectory data, and chose not to perform a quantitative intercomparison to other sea-ice drift data records like IFREMER or NSIDC, if this is what you call for with "The lack of comparisons with existing sea ice motion products would be relatively strange."

We indeed do not believe such initial data-description papers should hold quantitative inter-comparisons to existing products. We think the user community deserves better, and that product intercomparisons are best led by authors / teams independent from the data producers. Otherwise there is always the risk that the selection of reference data, metrics, reprojection, etc… favor one's product. The community is better served with independent and transparent product intercomparison studies, reported in a separate manuscript (e.g. Sumata et al., 2014; Sumata et al., 2015; Wang et al., 2022).

We note that neither the manuscript introducing the IFREMER data record (Girard-Ardhuin and Ezraty, 2012), nor that introducing the NSIDC data record (Tschudi et al., 2020) held an intercomparison to other products.

Sumata, H.; Kwok, R.; Gerdes, R.; Kauker, F.; Karcher, M. Uncertainty of Arctic summer ice drift assessed by high-resolution SAR data. J. Geophys. Res. Ocean. 2015, 120, 5285–5301.

Sumata, H., Lavergne, T., Girard-Ardhuin, F., Kimura, N., Tschudi, M. A., Kauker, F., Karcher, M., and Gerdes, R. (2014), An intercomparison of Arctic ice drift products to deduce uncertainty estimates, J. Geophys. Res. Oceans, 119, 4887– 4921, doi:10.1002/2013JC009724.

Wang, X.; Chen, R.; Li, C.; Chen, Z.; Hui, F.; Cheng, X. An Intercomparison of Satellite Derived Arctic Sea Ice Motion Products. Remote Sens. 2022, 14, 1261. https://doi.org/10.3390/rs14051261

To summarize our answer to your comment: in addition to our extensive validation against buoy trajectories, we will strengthen the comparison to NSIDC and IFREMER in terms of main characteristics (in section 4.3), but we suggest not to perform and report on a quantitative intercomparison against other satellite-based products in this paper.

In dialogue with the Editor, we address your suggestion as follows:

- We organized the existing text describing our CDR to others like NSIDC and IFREMER into a dedicated section (Sect 5 : Comparison to other CDRs).
- We added a new sub-section (5.3 Drift Duration) discussing the differences in drift duration (1 day for OSI SAF and NSIDC, 2- and 3-days for IFREMER).
- We added a Table to summarize the noted differences (Table 9).
- We added a new subsection (5.4 Accuracy against buoys) in which we compare the validation statistics we obtain against buoys, to what earlier authors have reported. This is not a new intercomparison done by us, but hopefully gives an idea of how our data validates against buoys relative to other CDRs.
- We placed the text providing an R&D outlook for the OSI SAF data record in a new section (Sect 6.4 Outlook for the OSI SAF CDR).

We hope that this re-organization of existing and new material will provide the reader and user with a good understanding of the characteristics and qualities of the OSI SAF CDR in the context of existing data.

5.    Section 4.3.4 mentions that for the free-drift model, the authors' method does not capture the multi-year interdecadal variability of the relationship between wind and sea ice motion vectors, which is a non-negligible problem for summer sea ice motion. I would like to know how this problem is handled by other products that have been applied besides the mentioned solution by Brunette et al., (2022). You only seem to mention that the NSIDC product use a constant wind-ice-ocean transfer coefficient.

To our knowledge, the NSIDC product is the only other CDR providing summer sea ice drift. IFREMER stops during summer. KWOK was winter only (and is no longer updated). We will investigate and write a short summary of our findings re: summer sea-ice drift.

Our investigations confirmed that only the NSIDC (and now the OSI SAF) CDR have coverage during summer. This is now captured with new text in Sect. 5.1 and in the new Table 9.

For completeness we mention that the dataset known as KIMURA (Kimura et al., 2013) also covers the summer seasons, however the data is not freely available and is not regularly

updated (to the best of our knowledge) and is thus not considered in our manuscript. Also, SAR-based sea-ice drift data such as from RGPS and from DTU-Space have vectors during the summer period, but these are not global multi-decadal CDRs.

Minor comments:

1.  "The first/second part of the period" appears several times in the manuscript, e.g., L631, L632 and L640. It would be more appropriate to clarify the period in which the product showed a large bias when reminding users of the precautions (Section 4.3.4).

We will revise. This was also pointed out by the 2nd Reviewer.

We reworded to remove these occurrences.

2.  L222 "The model is expected to be less valid in winter than in summer, and less in the Arctic than in the Antarctic, because of neglecting internal sea-ice stresses." This statement is a bit difficult for me to understand, please explain it in more detail.

We will revise and potentially split it into two sentences.

We reworded the sentence as: "Because it neglects internal sea\=/ice stresses, the model is expected to be less valid in winter than in summer, and in the Arctic than in the Antarctic."

3.  L412 I tried to find the Appendix B but couldn't.

We were referring to Appendix B of Stoffelen 1998. We will check with the editorial team if our citation format (Stoffelen, 1998, Appendix B) could be written differently.

We use the standard LaTeX command \cite[Appendix B]{stoffelen:1998} and assume it is rendered correctly.

4.  L508 Given that the multi-oi summer sea ice drift product is obtained by merging only wind-driven product, their RMSEs are exactly equal rather than very similar by comparing Tables 7 and 6.

Indeed. We will revise this sentence.

We changed the sentence to "In summer, the multi-oi RMSEs are equal to those obtained with the wind-driven drift product, since the latter is the only contribution to the multi-oi fields during summer."

**Reviewer comments RC2**

Review of

A Climate DAta Record of year-round global sea ice drift from the EUMETSAT OSI SAF

by

T. Lavergne and E. Down

Summary:

This paper introduces a long needed alternative to the NSIDC sea-ice motion data set based on satellite microwave radiometer observations. Albeit shorter, i.e. for 1991-2020, the description of the data set itself, how it has been derived from a well-established high-quality inter-sensor calibrated microwave brightness temperature data set and ERA5 near-surface wind speed data input to a free-drift sea ice model, and how the data have been evaluated with buoy observation is very good and provides a quite comprehensive insight into this new data set. The data set is available for both hemispheres at daily temporal resolution. While the data set is year-round only the winter months are based solely on satellite observations. Summer months data are solely based on free-drift sea ice model forced with ERA5 near-surface winds tuned to summer sea-ice motion derived from the latest generation of satellite microwave radiometer observations. Transition months are a blend. An essential detail to know is that the product contains sea ice displacement together with position and temporal information; the product does not contain readily derived sea ice drift velocity. The data set itself follows latest CF-conventions, is very-well documented in the netCDF files' global attributes and contains a useful set of flag values. The data are accessible via tools such as ncview and the Climate Data Operators (cdos), documenting their usability.

We thank the reviewer for their positive appreciation of our CDR and manuscript. We will address the sea-ice displacement / velocity aspect (among others) later in this review.

General comments:

GC1: There are a few aspects detailed in my specific comments that require a bit more information and/or rewriting parts of the text. To this belongs usage of different frequency channels for SSM/I / SSMIS versus AMSRX, the transition of sea-ice model based sea ice velocity to sea ice displacement, and an even more comprehensive motivation of and emphasis on the fact that the product contains displacement along the x- and y-axis of the EASE grid used - which makes an easy uptake and usage of the product not straightforward for quite a fraction of users. Also not clear is why on the one hand summer sea ice motion is derived from the free-drift sea-ice model but on the other hand the summer sea ice motion data are considered good enough for the required tuning.

GC2: I am not convinced that the discussion added regarding apparent differences in the bias been buoy and satellite (or multi-oi) data is mature enough to be kept in the form as is in this manuscript. What I am missing instead and was hoping that it would be touched somewhere in the discussion is the consideration of the free-drift assumption during summer. How "accurate" are the summer sea ice displacement estimates for sea ice concentrations below, say, 80%, or for almost closed sea ice conditions?

Both noted. We will address these issues as replies to your Specific comments below.

Specific comments:

L72: It seems you decided to use the near-90 GHz channels for your product because of their comparably fine resolution. But what about the weather influence at this frequency

which is known to be considerably larger than at the other channels offered by SSM/I or SSM/IS?

In fact this is contrasting what you write later-on about AMSR-E/2 where you explicitly state that you are NOT using the 89 GHz channels because of the larger atmospheric influence. This inconsistency needs to be clarified.

*We agree this needs to be clarified. We will add a short section 2.1.3 Choice of microwave frequency to explain why we used near-90GHz imagery for SSMIs and 36.5GHz imagery for AMSRs. Shortly, the choice of microwave frequency is a compromise between the level of details in the imagery (mostly driven by spatial resolution) and the stability of the imagery from one day to the next (better with channels that are less sensitive to the atmosphere). There are no perfect channels for sea-ice drift on these missions: this is the result of a compromise.*

We added a new section 2.1.3, moving some text from section 2.1.2:

*2.1.3 Choice of microwave frequency*

*We thus select the near-90 GHz imagery channels for SSM/I and SSMIS, but the 36.5 GHz imagery channels for AMSR-E and AMSR2. This choice is the result of a compromise between the level of details in the imagery (better with higher spatial resolution, thus higher microwave frequency) and the stability of the imagery from one day to the next (better with channels with less sensitivity to the atmosphere, thus lower microwave frequency).*

*For the SSM/I and SSMIS mission, earlier investigations concluded that the near-90 GHz channels offer the best compromise (Lavergne et al., 2010) as lower frequencies have much coarser resolution.*

*For the AMSR-E and AMSR2 missions, the 36.5 GHz channels offer the best compromise. The 89 GHz channels have higher resolution, but are also more affected by atmospheric liquid water path and surface melting. The 18.7 GHz channel has previously been found useful for sea-ice motion tracking during summer (Kwok, 2008) but our experience preparing the CDR is that the 36.5 GHz imagery provided at least as good results when compared to buoy trajectories (not shown).*

L92: "free drift" requires sea ice concentrations below about 80%. What is the uncertainty / bias in case this constraint is not met? Is this discussed and if yes in what context?

*We agree this should be discussed in our manuscript, e.g. in section 3.3 Summer sea-ice motion : wind-driven free-drift model. It is correct that the free-drift equation is derived from the general momentum balance equation under specific assumptions, including that internal sea-ice stresses are negligible, which can only happen if SIC is not too high (e.g. below 80%). In our case however, we use the equation of the free-drift model only as a parametric formula for sea-ice motion, and we tune the parameters A, θ and Uwg against data. The values for these parameters are not derived from the free-drift equation theory (involving the Nansen and Rossby number, the Coriolis constant, etc…). Because we only use the free-drift equation as a model to fit data, the assumption is that our tuned parameters will compensate for the theoretical limitations of the free-drift model (to the best of their capacity). We will add some text along those lines in section 3.3 Summer sea-ice motion : wind-driven free-drift model.*

We added the following paragraph to section 3.3:

The free-drift equation Eq. (1) derives from the general momentum balance equation under specific conditions, including that internal sea-ice stresses are negligible (Leppäranta, 2011). Strictly speaking, this can only happen if the sea-ice concentration is rather low, e.g. below 0.8. However, we apply the free-drift model to conditions with much larger concentrations in this CDR. This is because we use the free-drift model only as a parametric formula for sea-ice motion, and we tune the parameters |A|, θ, and Uwg against drift data. The values for these parameters are not derived analytically from the free-drift equation theory (involving the Nansen and Rossby number, the Coriolis constant, etc). Because we only use the free-drift equation as a model to fit data, the assumption is that our tuned parameters compensate to the best of their capacity for the theoretical limitations of the free-drift model.

L134/135: "in particular ... and 24 UTC" --> I have a problem understanding this. Every point in the polar hemisphere is covered by a number of satellite overpasses every day - those closer to the poles more often. I can imagine that for some regions there are overlapping overpasses at (roughly, don't count me on the full hours please), for instance, 4 UTC and 6 UTC and then later on at 15 UTC and 17 UTC (i.e. four overpasses in total) whereas there are other regions where these overpasses occur at 9 UTC and 11 UTC and later on at 20 UTC and 22 UTC. Does your weighing scheme mean that in the first case overpasses at 15 UTC and 17 UTC would given more weight than the other two (because of being closer to 12 UTC) while in the second case it would be the overpasses at 9UTC and 11 UTC? Doesn't this result in a preference of using TB values always from the same time of the day since overpass times are rather stable - within certain bounds? It seems to me that this results in regions where gridded TB values are predominantly either from the ascending overpasses or from the descending ones. Is this the intention?

Your understanding is correct. The spatial patterns of the mean observation times from these satellite missions are rather stable, but not constant because the orbits do not repeat exactly (in terms of UTC hours). You can see the result of the patterns on Figure 4, and note that the differences of observation time from one day to the next (right-most panel) is not uniformly 24h especially at lower latitudes.

We did not update the manuscript.

L136-138: "we compute ... is useful ..." I have two comments here.

- 1) Does the mean observation time use the same weighing as introduced above? If not my first case from the previous comment would result in 10:30 UTC and the second case in 15:30 UTC - which would not reflect the time of those swaths that has been given more weight than others.

Yes. We will modify the sentence to read: "In addition to the daily gridded TB, we compute the mean observation time for each pixel in the gridded image using the same temporal weighting".

We changed the sentence to read: "In addition to the daily gridded TB, we compute the mean observation time for each pixel in the gridded image using the same temporal weighting".

- 2) You state that the users of the CDR and your own evaluation activities would need this mean time. I agree. You might want to state for what and hereby stress that your CDR does not include any speed information but is solely about the displacement.

Yes. We can do that. For example: "This mean observation time can be useful for the users of the CDR to compute sea-ice velocity vectors (Sect. X.X) or compute model-equivalent displacement vectors, and we use it when collocating the CDR with buoy data (Sect. 3.6)".

From your several comments on this topic we will add a dedicated section (Sect. X.X) to discuss the sea-ice displacement / velocity topic.

We changed the sentence to read: "This mean observation time can be useful for the users of the CDR to compute sea-ice velocity vectors (Sect. 4.2) or to compute model-equivalent displacement vectors, and we use it when collocating the CDR with buoy data (Sect. 3.6)"

- This puts the question in my mind whether, when doing the consistency checks of your data set, investigated whether the time difference between two consecutive displacement maps provides a reasonable ice motion speed estimate when used to compute the velocity.

In each product file, we add variables holding the start and end observation times of the displacement vectors, so that users can compute the actual duration of the displacement (and hence the velocity).

We did not update the manuscript.

L142/143: The influence of this Laplacian filter seems to be quite delicate because on the one hand it removes gradients in the intensity while on the other hand it is supposed to enhance patterns. As written this seems contradictory and I invite you to provide sample information about how such a gradient might look like and where it comes from (i.e. what is its cause), and how one can discriminate between a gradient and a pattern.

Thank you. We use the term "gradient" for large-scale changes in the image intensity level, as can typically be caused by atmospheric effects such as water vapour. We use "patterns" for short-scale variations of the image intensity level that are stable in time and can be tracked by the motion algorithm. We will add some text explaining better what we mean here.

We changed the sentence to read: "The filter acts on two scales. First, it dampens large-scale intensity gradients across the images as well as intensity differences between the start and stop images as can be caused by passing weather systems. Second, it enhances small-scale intensity patterns in the image and stabilizes them in time, in view of their processing in the motion tracking algorithm".

L145: "rotated onto the EASE2 ..." --> It is inherently recognizable that you thereby switch from the meteorological convention of directions to the "grid world" and are only interested in the components along the x- and y-directions of the grid.

I guess this is the right moment to comment that: If you would have decided to provide a

sea-ice motion instead of a sea-ice displacement product, then this step would have been obsolete. I haven't found yet the motivation why you prefer to derive a sea-ice displacement product. I fear that the uptake of such a product by the user community won't be straightforward - particularly in light of the NSIDC sea ice motion product providing motion in x- and y-direction of the grid they used and the modified version of this product offered by ICDC which comes up with the u- and v-component of the ice motion in meteorological notation - an, to my opinion, much more handy product as it immediately represents the action of cyclones on the sea ice. In your product, one does not only need to play around with the time information to derive the motion (and with that to be at the same level as the NSIDC product) but one also needs to be very careful in the interpretation of the data as a positive x-displacement means eastward drift just north of Svalbard, southward drift in the Laptev Sea, and westward drift north of Bering Strait ... complicated ... and error-prone for a user.

This is a very good comment, and we must address it in the revision of the manuscript (and potentially in a later version of the CDR). We foresee the following changes will be implemented in the revised manuscript:

- We will add a new section (Sect X.X, possibly 4.2) to justify why we are focused on the dX and dY components of the displacement vectors (in short, because they are what the motion algorithm primarily retrieves) and why we refrain from giving easy access to velocities or other derived quantities such as speed and direction (in short, because we want to nudge the user to realise what satellite-derived sea-ice drift products really are, and because some transforms actually introduce pseudo-biases). This will borrow text from today's Sect. 3.6.3 "Validation Metrics" but make it more general.
- We will explain to the users how they should proceed for computing u/v velocities (in the meteorological convention) from our dX and dY. They do not need to rotate the dX and dY components: they can use the lat1 and lon1 variables from the product file. This will be a useful addition to the manuscript.
- We will provide a (python) notebook example in the accompanying github repo to demonstrate the conversion from dX,dY (km) to u,v (m/s) and refer to the notebook from the text.

We authored a new section (Sect. 4.2 Vectorial representation of sea\=/ice drift) to justify our choice of focusing on the dX and dY components in the paper and in the product files. This text also touches on the difference between velocity and displacement. This new section borrowed heavily from Sect 3.6.3 (Validation Metrics) that we decided to remove Sect 3.6.3 (it was too short).

Acknowledging that some users will want to access velocities and/or other components, we prepared a notebook demonstrating how these transformations can be done from the OSI SAF CDR (https://zenodo.org/record/8315156).

L183-189: Since your product is a sea ice displacement it would be good to provide a number for the maximum displacement in addition to the 0.45 ms-1. That way a reader can go back to the product and eventually check how far away a particular value in the displacement maps is from this maximum displacement value.

We will consider this. But since the actual duration of the displacement varies from place to place and during the day, it is not clear a single number is useful. We argue that the user should rather compute the speed associated with a displacement vector and compare it to 0.45 ms-1.

We did not update the manuscript.

L202-203: "value less than rho = 0.5" --> It seems that this is the first time you mention a threshold for the correlation. How about in all the other cases, i.e. before the rogue vector filtering: Is there a minimum value of rho which needs to be exceeded or will also displacement vectors with a rho = 0.35 (as an example) make it into the product?

We will add more information about the correlation thresholds for the other cases. All this information is in the product ATBD, but we can bring some of this here.

We added the following sentence at the end of 3.2.1: "Vectors retrieved with a maximum cross-correlation of less than 0.3 are discarded and the product grid has a missing vector."

L218: "... during summer" --> I suggest to somewhere in this paragraph add information that surface melting reduces the identification of patterns in the observed TBs that are required for the tracking; one classical example applying to the Arctic is the loss of the radiometric difference between FYI and MYI. Without this additional information a reader might think this lower accuracy is merely due to melt pond coverage.

Good idea, we will add such sentence.

We modified the existing sentence to read: "… is much less reliable because of surface melting *(e.g. melting of the snow cover, reduction of the difference in emissivity between first and multiyear sea-ice)*" (the text in italic was added).

L224: I note that you switch from displacement (during winter) to velocity (during summer) here.

Indeed. We will add a sentence to specify that the free-drift equation is introduced as velocities, but that we apply it to yield displacements in the product files.

We added a sentence to section 3.3.2: "Sea-ice velocity vectors are finally converted to sea-ice displacement vectors by mutiplying the components by 24h."

L234-236: I have two questions here:

- 1) You stated that during summer the CMCC algorithm is not suited well to derive ice motion (displacement vectors). But here you use maps of ice motion vectors from July derived nevertheless with the CMCC for the tuning. This needs to be explained better as it seems contradictionary.

Agreed. This will be stated either here, or in the discussion section related to the summer sea-ice motion bias (Sect. 4.3.4).

We have now added the following text at the beginning of Sect. 3.3 *Summer sea-ice motion : wind-driven free-drift model* to clarify our approach for the CDR.

*More recent missions like AMSR-E and AMSR2 offer lower microwave frequency channels at a better spatial resolution which allows some level of accuracy for motion tracking during summer but a) they are only available after 2002 and b) the resulting daily fields can have many gaps over melting sea\=/ice. To offer a consistent CDR over 30 years, we do not distribute the motion vectors retrieved from the AMSR\=/E and AMSR2 missions during summer, but rather use them to tune a free\=/drift model. We then use the free\=/drift model to prepare daily sea\=/ice drift fields during summer, that we distribute as part of the CDR.*

We also added the following sentences at the end of *Sect. 6.3 Outlook for the OSI SAF CDR*, discussing next R&D steps towards future versions:

*Another approach would be to use the vectors derived from the AMSR\=/E and AMSR2 missions as input to the merged product during summer, instead of using them only to tune the free\=/drift model. This would however require careful consideration of the temporal consistency of the CDR when satellite\=/based summer vectors are introduced at the start of the 2000s.*

- 2) You state that you use data of years 2002 through 2020. Did you perform the tuning individually for every year? Or did you average over data of all June, all July, and all August months of the years 2002-2020, respectively, to arrive at one set of tuning parameters per month?

We do the parameter tuning per months, using all available years in the tuning. In the end we obtain one set of parameter for January (using all January months from 2002 through 2020), one set of February (using all February months from 2002 through 2020). We agree this is not well explained and we will clarify in the text.

We clarified the text to read "Parameter-tuning is performed on a monthly basis, using all available years (2002-2020). We thus obtain 12 monthly maps of parameters |A|, θ, and Uwg".

L240-242: I agree that during the earlier decade of the CDR the sea ice covered more area during summer and I welcome the thought to be able to apply the wind-driven model also in the early decade of the CDR. However, I am wondering whether the extrapolation isn't potentially causing artefacts that negative influence the tuning, and whether without this extrapolation the tuning wouldn't be as good as with. After all, all you need is a representative set of conditions relating sea ice displacement observed to ERA5 near-surface wind vectors and I would assume your AMSRX period is long enough for this purpose.

The tuning itself is not influenced by the extrapolation. We first perform the tuning in areas where we have enough sea-ice drift vectors, then perform an extrapolation towards the areas where no tuning could be performed. That being said we could have used different gap-filling strategies than the extrapolation from neighbours, e.g. filing the gaps with fixed mean parameter values. We have too strategies to avoid serious impact by extrapolation

artefacts: 1) we performed visual inspection of the parmeter maps (there are only 12 maps, for 2 hemispheres); 2) we have a different status_flag for vectors derived from the extrapolated parameters.

We will add a sentence or two about the extrapolation and its potential artefacts.

We added a sentence: "We underline that the extrapolation is performed after the parameter tuning and cannot degrade its performance." We also mentioned visual inspection in the following paragraph: "The *monthly* maps of parameters (...) *are inspected visually*, and finally saved …"

L243: "small radius" --> consider providing a value. How small?

Yes. The value is in the ATBD but we can bring it in the paper.

We changed the sentence to read: "Finally, the maps of parameters are smoothed using a Gaussian weighting filter with *a 62.5~km sigma*"

L247 / Fig. 2: It might be a trick of my eyes but when I compare the map with the legend I would say that the turning angle is merely between -25 and -40 degrees to the left; I can hardly see any values around (just) -20 degrees.

L256 / Fig. 3: Also here I would be inclined to see mostly larger values than 20 degrees. Table 3 actually confirms my view - also the one noted in L247.

We will double-check our colour scales and text. The mean values reported in Table 3 indeed tend to confirm your impression.

We double-checked our colors scales and text and updated both to better reflect the data. The values in the maps were indeed larger than what we were indicating.

L256: "with generally thinner sea ice" --> I am wondering whether the overall lower SIC and with that better match between the free-drift assumption and the actual sea ice conditions doesn't play also a role here - perhaps even more than the thinner sea ice?

This might well be the case and we will add this as a possible explanation.

We changed the sentence to read: "which is compatible with a generally thinner *and less compact* sea-ice cover." (change in italic).

Figures 2 and 3:

- If possible I would have these figures to extend across the entire text width such that one does not need to zoom 200% or more to see what is in the figures in detail.

Agreed. We will enlarge them for the revised manuscript, and hopefully they will also be bigger in the final edited file.

Figures were enlarged.

- Are these figures showing an individual year or are we looking at a July mean for 2002-2020?

They show the July parameters, obtained from using all July data in the period 2002-2020 (see our answer to your earlier point).

No change to the manuscript.

- You might want to explain the white disc in the center of the maps shown in Figure 2.

Yes, we will. This is the imprint of the polar observation hole. The interpolation / extrapolation discussed earlier will fill this region so that the free-drift parameter can be applied there as well.

We added the following sentence in the caption to Fig. 2: "These maps show the parameters before the spatial extrapolation. The white pixels around the North Pole and along the coasts are filled in the extrapolation step."

- You might want to explain the white fringe around the coastlines.

Yes, we will.

See our answer to the previous point.

- Please add units for the turning angle and the under-ice ocean current; I note in this context that you could harmonize the name between the title underneath the panel and the caption.

All good suggestions, we will implement them.

Fig. 2 and 3 : the figures were improved, units were added, and the labels were harmonized with the caption and text.

L271/2 72: "we always compute two ... of the month" --> please try to give more details here. There are many ways how to interpret your writing:

1) You only compute two sea-ice motion fields at the very end / beginning of the month, i.e. June 30 vs. July 1, July 31 vs. Aug. 1

2) You compute two fields for every day of the summer months and then interpolate linearly between maps of June 1 and July 1, June 2 and July 2 and so forth.

3) Only within a certain period of time you blend the two motion fields, e.g. June 26 with July 1, June 27 with July 2, ..., June 30 with July 5.

Indeed, this requires a clarification. To ensure a smooth transition across the month transitions, we always compute two sea-ice velocity fields based on the parameter files from the "bracketing" months. These two velocity fields are then blended by linear interpolation, weighted by the difference between the simulation date and the mid-month dates of the

parameter files. For example, for a free-drift simulation for June 21st, the two bracketing months are June (16th) and July (16th).

We will clarify this in the text around L271, but also when we introduce the monthly parameter files at the end of Sect. 3.1.1 "Tuning the free-drift model".

We replaced our sentence L271/272 with "To ensure a smooth transition across the months, we always compute two sea-ice velocity fields based on the parameter files from the two "bracketing" months. These two velocity fields are then blended by linear interpolation, weighted by the difference between the simulation date and the mid-month dates of the parameter files. For example, for a free-drift simulation for June 21st, the two bracketing months are June (16th) and July (16th)."

L273: "sea-ice mask" --> could you remind the reader where you take the sea ice mask from?

Yes. We can note that we take it from the OSI SAF Sea Ice Concentration CDR v2 and cross-ref to Sect. 2.3.

We added the sentence: "This mask is from the EUMETSAT OSI SAF Sea Ice Concentration CDR v2 (see Sect. 2.4)."

L280: "gapfill whole winter days ..." --> While I can understand that you blend shoulder month ice motions and solely use the wind-driven product during summer I have my difficulties to understand why you gapfill the satellite product with these model data during winter, a time when the free-drift assumption often does not hold. This seems to degrade the quality of the CDR during winter. Yes, you do provide a flag for these grid cells but I would assume that 90% (at least) of the users see a gapfilled product and don't care what the source is and whether they should potentially exclude these grid cells because these are not based on the satellite observations.

Users have different levels of expertise, and different levels of applications. It is a difficult task to address them all with a single CDR. Our approach has been to provide a gap-free CDR as an entry point for all (non-expert) users with a multi-source / merged dataset. There we use the quality flags, and higher uncertainties, to warn the user. We also write user documentation and this publication. Given that sea-ice motion is very variable from day to day, mostly due to wind forcing, we think it makes more sense to use the free-drift model than other interpolation methods (e.g. a linear interpolation from neighbouring days). We underline that winter free-drift estimates are only to fill whole days at a time (not grid cells here or there which are interpolated from neighbouring vectors). We will try to strengthen this message in the text.

As we already note, expert users can access the flags and throw out the winter free-drift data, and can use the time series from individual satellite missions to do their own application.

We modified the last paragraph of Sect. 3.4.3 to read: "Wind-derived motion fields have sometimes to be used during the winter season, when missing satellite imagery leads to a lack of satellite motion vectors. The wind-derived motion field is then used directly in the

multi-source product. The dates at which this occurs are listed in Appendix A, and a specific status_flag value is used in the product files. Because we aim at a gap-free CDR, we thus apply the free-drift model in winter, despite it being theoretically less accurate because of larger internal stresses. Given that sea-ice motion is very variable from day to day, mostly driven by wind forcing, we deemed it makes more sense to use the free-drift model than other interpolation methods (e.g. a linear interpolation from neighbouring days). We underline that winter free-drift estimates are only to fill whole days at a time (not grid cells here or there which are interpolated from neighbouring vectors). Users who do not want to use free-drift estimates during winter are invited to use the status_flag."

L297: "drift vector" --> just because it strikes me in this moment: I am not sure the reader fully understands why sometimes "drift" and sometime "motion" is used. I am wondering whether, for the sake of clarity, you at some point in the introduction specify what you mean by motion and drift, and that these two terms can be used interchangably. Ideally, though, you switch to "displacement" where you refer to a distance traveled by the sea ice (as is actually the variable in the product) and to "velocity" where you refer to the speed with which the sea ice displacement has taken place (as is the output of the wind-driven model). I could imagine that this could avoid confusion reading the paper. At least I needed to re-assure myself whether in a particular passage you were writing about the displacement or velocity when using drift or motion.

This is a good point. We use motion and drift interchangeably and will make it clear in the introduction. Displacement and velocities are not the same thing and we will also note it early in the text. We might even create an Appendix "Terminology and disambiguation" if we find that these explanations (together with those why we prefer dX/dY to u/v) take too much space.

We added the following sentence at the end of the Introduction: "In this manuscript, we use terms sea-ice drift and motion interchangeably to refer to the fact that sea-ice moves. We however make a distinction between sea-ice displacement vectors, velocity vectors, and sea-ice speed (among other terms) as explained in Sect. 4.2."

Sect. 4.2 is the new section already introduced in an earlier reply.

L300/301: "adjusted to ... Sect. 3.5.)" --> I did not find a sufficient description about how this adjustment is carried out; Section 3.5 is rather short as well.

Noted. We also agree that Sect. 3.5 will be extended.

We extended Sect. 3.5 introducing a new equation for how uncertainties are adjusted for the time-mismatch to 12utc. The new text reads (see track-changed PDF for a better formatting).

*The uncertainty values $\sigma_k$ introduced above are valid when using the single-sensor ice drift vectors with their associated maps of start and end time provided in the product files. As shown later, these can easily vary between 8 UTC and 16 UTC across the product grid depending on the orbit and instrument characteristics (see Fig. 4). When rather using the vectors as if they were from 12 UTC and 12 UTC, the uncertainties must be raised. We compute the raised uncertainty $\sigma_{12utc_k}$ with a 2nd order polynomial formula:*

*$\sigma 12k\ (\delta t) = a \times \delta t^2 + b \times \delta t + \sigma k$ (3)*

*where $\delta t = |t - 12\ UTC|$ has units hours. The values of coefficients a = 0.015 and b = - 0.005 were obtained by running a validation experiment where we deliberately collocated buoys and satellite product with wrong time information, allowing us to explore the increase in uncertainties with an increase in time mismatch (from -10 hours to +10 hours). This exercise was conducted with the near-real-time OSI SAF sea-ice drift product, and was not repeated for the CDR as we believe the coefficients apply.*

*We compute maps of $\sigma 12utck$ using Equation (3) for the single-sensor products that are input to the multi-sensor merging step (see Equation (2) in Sect. 3.4). The uncertainty values of the merged multi-source product are computed by combining the uncertainty values of the single-sensor products $\sigma 12utck$.*

L314 / Section 3.4.3: What I am missing in this section is how the wind-driven model ice motion vectors = velocities will be transferred to displacement vectors so that these can be merged with the satellite product.

This is a good point. We actually use the free-drift model to generate maps of sea-ice 24h-mean Eulerian velocity vectors, then multiply them by 24h to obtain the displacement vector. This results in an Eulerian displacement while the satellite-based displacement vectors are Lagrangian displacements. We will attempt to clarify this in the new Sect 4.2 where the choice of dX/dY over u/v is justified.

We added the following sentence: "Sea-ice velocity vectors are finally converted to sea-ice displacement vectors by mutiplying the components by 24h (see a note about this conversion to displacement in Sect. 4.2)."

L319-322: "In the spring month, ... nominal at the end." --> I am wondering whether it would make sense to share typical nominal values of the uncertainties with the reader here. It would be good to learn whether this ramping begins at 0.5 km, 5 km or even closer to the 10 km stated. I also note that in case of the wind-driven product the status - at this point of writing - is that these are sea ice velocities and hence a standard deviation of 10 km does not apply properly.

This is a good point. At this stage, the reader does not know what is the typical scale of the satellite- or wrind-based uncertainties since these derive from the validation results that are introduced later in the manuscript. We will do as you suggest and mention typical values here.

We added some text in a sentence: the uncertainties of the CMCC-calculated drift vectors are at their nominal values *(typically 2.5 to 3.5 km standard deviation in the northern hemisphere, 3.5 to 4.5 km in the southern hemisphere, see Sect. 4.3)* at the start of the month, and at a high value (10 km standard deviation) at the end of the month.

L326: "The wind-derived motion field ..." --> interesting to see the effort you carry out to have a smooth transition across the shoulder months from winter to summer and back to winter while here there seems to be no further effort to establish a smooth transition - at least not from your writing; perhaps this is in the ATBD?

There are no further efforts in that case. We could have re-used a smooth transition in case of winter gaps but the risk would have been to degrade high-quality satellite-based winter drift vectors with lower-quality wind-based vectors around the gaps. We will note that this is an abrupt process, with no transition.

We added the following sentence: "No smooth transition is implemented from the day before or after to not degrade high-quality satellite-based sea-ice drift vectors with lower quality winter free-drift vectors: this is an abrupt replacement"h.

L333/334: "the grid spacing between vectors is 75 km" --> After the abstract this is the first time the grid spacing of the product is mentioned. This oversight should be corrected at an appropriate place in the methods section.

Yes. This will be added for example at the end of Sect 3.2.1 The continuous maximum cross-correlation method.

We added the following text at the end of Sect 3.2.1 : "For this CDR, the CMCC is applied on an EASE2 grid with 75 km spacing, every 6th imagery pixel (itself on an EASE2 12.5 km grid). Each vector is built using sub-windows of 11 x 11 satellite imagery pixels."

We also added a sentence for the wind-driven vectors in Sect 3.3.2 : "Since both the ERA5 winds and the parameter files are prepared on the EASE2 grid with 75 km spacing, the resulting wind-driven vectors are directly on the same grid as the satellite-derived vectors."

L349/350: "that increase ... to 12 UTC" --> this seems to be the part you are refering to from L300/301 but I don't understand what you are doing here. The method is not clear quantitatively and it is also not clear what you mean by "where the drift period of the single-sensor product is far from 12 UTC to 12 UTC". Clarification required.

Yes. This will be clarified and extended. In the ATBD we have a graph and a formula, but we did not want to bring this level of detail in the manuscript.

We extended this content with new text and formula (from the ATBD) to clarify what we are doing. See our answer to your earlier point.

L362: Why this step? A buoy that reports no displacement over a certain time period seems to be stuck with the sea ice it sits on somewhere. Removing those records between t_1 and t_2 with t_1 = begin of being-stuck-period and t_2 = end of being-stuck-period complicates to generate a product with certain temporal resolution, doesn't it? It is like removing SIC values in a SIC maps where the values to not change from 100% (or 0%).

What happens if one wants to get the ice drift information for that particular buoy for a day that lies between t_1 and t_2? I assume that date might not be found then.

This should have been formulated differently. Our test was to capture unphysical data along the buoy trajectories, e.g. when data records showed flickering between fixed positions. Our test would not have discarded physically realistic slow-moving buoys. We will clarify.

The list item L362 was changed to "- remove portions of trajectories where the position flickers between fixed positions."

L363/364: Does this work for buoys that have very variable drift speeds, e.g. being set out on moderately fast drifting sea ice in the Beaufort Sea end of summer, then being incorporated into perennial sea ice for 2 years with very slooooooooow movement and then entering the transpolar drift stream being exported through Fram Strait? Would particularly the observations in the latter region mostly being discarded? Showing examples might help.

At the side: How do you adequately compute the standard deviation of the discplacement when you, as described in the previous step, remove positions where the buoy (safely) did not move?

We agree that this test could have removed trajectories with high variability during the lifetime of the buoy. We chose 3 times the standard deviation by inspecting the trajectories that were detected and concluded that this was an effective threshold to mostly detect rogue positions and not physically realistic trajectories. In future versions we might rather adopt a test based on a moving average and standard deviation.

We will add a sentence or two to state how these quality-control checks can be improved in future versions of the validation exercise.

We added the sentence: " This test would possibly be more effective with a running window along the trajectory, which will be considered in future versions of the CDR."

L386: Is this "40 km" triggered by the grid spacing of the product of 75 km? Presumably you want to have reference and product vector to begin in the same grid cell and 40 km is then your approximation of half the grid-cell width of 37.5 km parallel to the x- or y-direction, right?

The spatial collocation uses the nearest-neighbour technique, so we are already ensured that the reference and product vectors are from the same grid cell. The additional threshold of 40 km is to bring a symmetry in the collocation to discard the corners of the grid cells. We could have used half the grid size (37.5 km) indeed.

No change to the manuscript.

Figure 4: I suggest to add in the caption that the start and stop times shown in the two left columns are given relative to 12 UTC? Otherwise I don't understand what the negative time denotes.

Agreed.

A sentence was added in the caption to Fig. 4 : "The start and end time are shown as a time-difference to 12 UTC".

L448/449: I am suggesting that - in a future release of this CDR - you switch to 2-sigma uncertainty values to be in line with the GCOS requirements.

This is noted. However most users will naturally use uncertainties provided in a product as 1-sigma, which is why we might keep it this way. But it is indeed important to keep in mind that GCOS's requirements are expressed as 95%-probability (2-sigmas) when comparing our uncertainties or validation results to the GCOS requirements.

No changes to the manuscript.

Figure 6:

- I note that the x-axis notation is showing "validation data" while earlier in the text when describing the validation you used the term "reference data". I suggest to stay consistent throughout the manuscript.

We will review the text and figure text and ensure consistency of the notations.

Implemented.

- Is there any pressing reason why the logarithmic scale shown changes with respect to the total range shown between the different panels? If this was on purpose you could note in the caption that the ranges differ. If this was not on purpose you could consider harmonizing the ranges.

We will note in the caption that the ranges differ.

This was noted in the captions of Figure 6, 7, and 8.

- There is this small notion "All Flags" in every panel. Would you mind to state in the caption and/or the text what you are refering to here?

Yes. We will either mention it in the caption or remove that mention.

- You could mention in the caption that these are full-mission results.

Yes, we will do this.

Implemented in the captions of Figure 6, 7, and 8.

Note that these comments apply to the following figures of the similar kind.

Noted, thank you.

Implemented in the captions of Figure 6, 7, and 8 (see above).

L483: "10-15 km resolution" --> Is it really the native resolution of the satellite imagery which counts here? Isn't it rather the 75 km grid spacing that is used?

It is indeed the native resolution of the satellite imagery we want to point out here. We think it is remarkable that the RMSE against buoy trajectories results in sub-image-pixel accuracy.

No change to the manuscript.

L525/526: "One exception seems to be ..." --> What does this mean and/or what do you want to state with this sentence? One could ask the question why this is not confirmed by dY and/or what went wrong with dX here?

We do not have an explanation for this different behaviour of the F14 dX RMSE. We wrote the sentence as a description of the figure. We might remove the sentence as it does not bring new information.

We removed the sentence.

Figure 9: Consider enlarging the panels in this figure. It is almost impossible to see that MULTI-OI summer is identical to ERA5; you could used dashed lines for MULTI-OI so that ERA5 "shines" through. Where are the buoys?

Yes, the panels will be enlarged, both for the revised manuscript and for the edited version. We will consider using dashed lines for MULTI-OI. It was an error in our legend to indicate the buoys : we are showing the RMSE against the buoys so that we can't show the buoys independently.

Figures were enlarged and updated (dashed lines for MULTI-OI, removed the "buoy" label).

L554: "the better of the SMMR period" --> What is this?

We will indicate the exact time period.

We indicated the period : daily imagery was only available for the first 20 days of the mission in Oct./Nov. 1978).

L569: "we compute drift vectors every six image pixels" --> you could emphasize that this applies to both x- and y-direction.

Yes, we will do this.

Implemented : "... we compute drift vectors every 6th image pixels *along both x- and y-axis …"*

- Stupid question: Did you try to derive the same ice motion fields using a different set of center (?) 12.5 km grid cells, i.e. moved by 2 grid cells in x and 3 grid cells in y-direction?

We did not do this.

No change to the manuscript.

- The (?) I put behind center was to remind me that with 75 KM grid spacing there is no real center pixel because of the even number of 12.5 km grid cells per 75 km grid cell. A problem?

The center points of the 75km grid fall exactly on the center points of the 12.5km grid so there is no alignment problem.

No change to the manuscript.

L616: "our wind-derived ... in the northern hemisphere" --> I am not that much convinced about this partitioning into two distinct periods. Actually 4 of the 5 largest (by magnitude)

biases fall into the AMSRX period. Also, while during the first decade positive biases are almost missing they occurred more often during the AMSRX period with values up to 0.3 km. I am therefore not convinced that the discussion as written and illustrated.

See the combined answer below.

L691/692: Again let me state that I am not that much convinced about emphasizing this first half / second half thing. Actually it is 11 years vs. 19 years, so the tuning is actually not just from one half of the 30 year CDR. I agree there are different biases but I don'T find the message is particularly clear. I would rather state the bias is quite variable and I am not sure whether all eventualities that might have caused this bias are adequately discussed. What do we know about potential trends in the ERA5 winds in the Arctic / Antarctic and/or how well (and with which data) has this parameter been evaluated in the polar regions?

For the two comments above : we will review our text and revise it with your input in mind. A bias or trend in ERA5 wind is indeed not something we can rule out. But in our revision we still want to keep the warning to our users to not use the summer data for trend analysis.

We have now reviewed our text for Sect. 6.2 "Negative bias for wind-derived motion (summer season)" to reflect your suggestions. In particular we have removed the reference into a first- and second-half, and changed our formulations to hypotheses rather than knowledge. We also mention that we cannot rule out other explanations such as a trend in polar ERA5 winds. The new paragraph reads:

*\Fref{fig:valid_timeseries_dx} (right column) however documents the negative summer bias in the \dX\ component (same for \dY, not shown). We have no definite explanation for this bias, nor for why it reduces in the later part of the CDR (2016\=/2020). We recall that the free\=/drift model was tuned against satellite\=/based sea\=/ice motion from the AMSR\=/E and AMSR2 missions, thus in the period 2002\=/2020 (\sref{sec:winds}). Firstly, validation of the summer sea\=/ice motion data from AMSR-E and AMSR2, not included in the CDR bu shown on \fref{fig:valid_timeseries_dx}, exhibits a larger bias than in the winter season. This bias in satellite\=/derived motion is amplified into the free\=/drift model parameters and the predictions from that model. Second, it is well documented that sea\=/ice motion has experienced a positive trend over the last decades, concomitant with sea\=/ice thinning in the Arctic \citep{rampal:2009:drifttrend,spreen:2011:drifttrend,kwok:2013:arctictrend,olason:2014:arctic drift} and stronger winds in the Antarctic \citep{holland:2012:antarctictrend,kwok:2017:antarctictrend}. Since we tune our free\=/drift model against data from the later part of the period, where sea\=/ice motion is transitioning to a faster regime, we might have difficulties to capture these trends in our summer drift. We do not have access to a good enough buoy coverage for the southern hemisphere to assess the temporal evolution of the negative summer bias there, but an overall bias is present (\tref{tab:stats:winds}). We finally note that we cannot rule-out that other factors, e.g. an (hypothetical) trends in polar ERA5 winds, contribute to the observed trend in the bias of our summer drift.*

Another suggestion I have with respect to future plans is whether a two step validation process wouldn't perhaps be more promising. I am thinking of using the buoy derived ice

displacement to evaluate SAR-based ice displacement vectors to have some spatial distribution of the reference data to compare the CDR with in a second step.

This is a good suggestion that might even help quantify the representativeness uncertainty between the buoy and the PMW sea-ice motion data. One issue with the SAR ice drift is that until recently with the availability of Sentinel 1A and 1B or RCM the repeat time of the imagery did not allow for many 24h drift vectors from SAR. It is not clear if we can use, e.g. ERS or Envisat SAR drift information from the 1990s and 2000s.

No change to the manuscript.

Typos / editoral comments:

L46: "mission" behind "AMSR-E" can be deleted.

Will do.

Implemented.

L62: It is correct that buoy data enter the product of Tschudi et al. However, this does only apply to the Arctic, not the Antarctic. You could consider adding this to your text.

Thanks. We will add this to the text. We might move this sentence following Reviewer 1's comment.

Implemented.

L78/79: Perhaps write a bit more about this data set? Are this gridded data? What is the temporal information?

We will expand. This is L1B data in swath projection, not gridded.

We added a sentence: "It contains quality-controlled and intercalibrated Level-1B (swath-based brightness temperature) data for SMMR and all SSM/I and SSMIS."

L88-90: Again it would be helpful to know how this data is provided (as swaths or as daily gridded data) and also information about the time sould be given.

We will expand. This is again L!B data in swath projection, not gridded.

*We extended the paragraph to read:*

*For both missions, we access brightness temperature data in swath projection (Level-1B data). We use the AMSR-E Level-1B data from the FCDR V003 (\href{https://doi.org/10.5067/AMSR-E/AE\_L2A.003}{10.5067/AMSR-E/AE\_L2A.003}) of \cite{aschcroft:2013:amsre}. We access the GCOM\=/W1 AMSR2 Level-1B data directly from the Japan Aerospace Exploration Agency (JAXA) G\=/portal.*

L100: "full-resolution" means what?

This is the terminology used on the IABP website "Full Resolution Data (typically hourly)". We will add the same information "(typically hourly)" in our text.

We added "(typically hourly)" in our sentence.

L103-106: "In addition, ... Derocher 2020)." --> Would it make sense to the reader to put this information right behind the sentence dealing with the Arctic buoys ending in L101? All additional data sources given here are from the Arctic and none are from the Antarctic - unless I am mistaken.

It would make sense, yes.

We implemented the suggestion and moved the sentence about Antarctic sea-ice buoys to the end of the paragraph.

L111: You could consider adding a short notion why the Fram Strait is excluded from the main validation and/or point to the respective later section wherein you deal with this issue.

Yes, we will point at the later section.

We added a reference to the appropriate later section.

L121: "Theoterical" --> "Theoretical"

Will do

Done

L214: "increase" --> "increased"?

Will do

Done.

L226: By "surface wind" you mean the 10-m wind?

This equation is quite general and can be expressed with both surface wind, geostrophic wind, and 10 m wind. The value of A and the turning angle would be different. We will consider change to 10 m wind since this is what we use in the end in this manuscript.

We changed to "10-m wind" twice in this section.

L247/248: "older and thicker sea ice" --> In L258 you rightly connect thicker sea ice with "a larger impact of neglected internal stresses". I believe it would be a good idea to mention this here as well.

Yes, we will do this.

Done. The sentence now reads "where older and thicker sea ice *with larger internal stresses* is".

L262: Please check the Greec letter for the turning angle. Also, there are two sentences beginnign with "Table 3 ..." which could possibly be merged?

We will change the Greek letter and avoid the repetition.

Both implemented.

Table 3: Just a minor editoral remark: Please check how you want to (should) write the respective hemisphere ... Northern Hemisphere or northern hemisphere ... I guess there are rules but I am not aware of these at the moment.

We will check.

We have now consistently changed to capitalized versions in the revised manuscript. This will be checked again at copy-editing.

L285: Consider again adding the information that the NSIDC product uses buoys in the Arctic only.

Yes, we will.

Implemented when the sentence was moved and reformulated in Sect. 4.1.

L292/293: "The mask over ..." --> It might make sense to move this sentence to behind the one ending in L290 with "86N)"?

Yes, we will move.

Done. The end of the sentence now reads "... 86N, to ensure that the check is consistent across satellite missions (with varying widths of the polar observation hole, Sect. 2.1)"

L325: "sometimes to used" --> "sometimes to be used"

Yes.

Done.

L335: "would never contains" --> "would never contain"

Yes.

Done.

L389-391: Here you use two times the expression "validation", in form of "validation vector" and in form of "validation data". The reader would possibly appreciate to have this new term explained or, alternatively, replaced by what applies: "product" or "reference".

Yes.

Done.

L402: "variables the" --> "variables of the"

Yes.

Done, but this part of the text was modified extensively.

L404: "product" --> "products"

Yes.

Done, but this part of the text was modified extensively.

L409: "he" --> "the"

Yes.

Done, but this part of the text was modified extensively.

L410: "We do to not report" --> please check.

Yes. Should have been "We do not report". But this text will be part of a new text Sect X.X discussing the dX/dY vs u/v choice.

Done, but this part of the text was modified extensively.

L487: "that that" --> please check.

Will do

Fixed.

L497/498: "The wind-driven ..." --> Please check this sentence for plural "s": "products ... seems ... it extends ..."

Will do

Done

Figure 7: What does the "(NONE)" in the first line of the text in every panel mean?

We will remove it, it does not hold any information in that case.

Done.

L598: "does ... results" --> "does ... result"

Will do

Fixed.

L653: "th" --> "the"

Will do

Fixed.

**Editor Comment:**

Two reviewers both find benefits to this data product, but also substantial issues. I invite you to continue to the next round. At ESSD, this is responding to review comments without updating the manuscript. If responses are deemed reasonable, then revisions.

Thank you for this invitation. We prepared replies to reviews (see above).

I note that CF compliance is mostly met but there are still two issues according to my compliance checker, and there are several missing "Highly Recommended" and "Recommended" fields for ACDD compliance. Please fix these.

Thank you. As you know, we already went through a round of revisions of our metadata after your initial feedback earlier in March. We did then fix the ERRORs and took on board some of the Highly Recommended or Recommended fields for ACDD compliance, but not all.

For example, the *cfchecker.py* compliance checker "highly recommends" standard_name for all variables, but some of the variables in our file have no registered standard name at CF. This is fine (in a CF sense): if a variable does not correspond to a registered standard name it should not use a standard_name attribute. Also, none of the possible ISO 19115-1 data types seemed to apply as coverage_content_type. We also do not think vertical datum is required for our 2D surface product.

All in all we think that our set of metadata is extensive, and that users will find the information they need. It is also too late for us to amend the metadata now that the data record was accepted by the EUMETSAT data portal, and was uploaded there.

We hope you agree that our metadata are suitable for publication of the paper in ESSD.

Regarding CF compliance, there is documentation for you to generate names that may not exist in the standard vocabulary: http://cfconventions.org/Data/cf-standard-names/docs/guidelines.html I note that if I drag-and-drop your product into QGIS it appears to work well, but there are minor issues: It reports "Unknown CRS". I recognize you don't want to update metadata but there are some things that can be improved. "Conventions does not contain 'ACDD-1.3'" can be fixed with, I believe, a semi-colon between the "CF" and "ACDD" strings in that field. For 'coverage_content_type' I believe one of "image, thematicClassification, physicalMeasurement, auxiliaryInformation, qualityInformation, referenceInformation, modelResult, or coordinate" might be imperfect but more useful than an empty field. Are you sure that "uncert_dX_and_dY" and "qualityInformation" does not fit? If not, then what exactly is "uncert_dX_and_dY"? While you're at it, it should be easy to improve the other recommended issues.

We thank the Editor for the guidance in improving the metadata of our data record. The :Conventions attribute was fixed and now reads "CF-1.7,ACDD-1.3". We added attribute :coverage_content_type = "qualityInformation" to the uncert_dX_and_dY variable. Other offending global attributes describing the time period covered by the product files where fixed.

Please change "1991 to December 2020" to "1991 through December 2020" (and similar phrasing elsewhere, e.g., "1979 *through* 2016" on L100) if the data includes December 2020. If it does not, then "1991 through November 2020".

We will correct this and similar phrasing.

Implemented.

I would also like to see the processing software workflow released that generated these products. When I download your software from Zenodo and try to run it, I get an error:

OSError:          [Errno          -90]          NetCDF:          file          not          found: b'https://thredds.met.no/thredds/dodsC/metusers/thomasl/SIDrift_CDR_v1pre/auxiliary_files/i nv_params_osi455_nh_200301-202012_1day.nc'

Which is OK! I'm looking for *open* science, not *reproducible* science. That being said, the software only seems to be code for generating figures. Please share your processing software. The actual work. This would let anyone who has any methods questions not answered by your manuscript find out the answer for themselves (for example, how map projection errors are handled).

For the OSError: we moved the CDR to its official location now, and we will update the github code with the new location when submitting the revised manuscript.

Done.

Releasing the processing software is already in our todo list from our interaction with you in March. We will have the processing software ready for final paper publication.

The processing software is not open at time of submitting our revised manuscript and this point-by-point reply. We are working on opening the processing software.

All figures: Bigger fonts!

Yes. We also hope that the figures will be bigger in the final manuscript format.

Done, we will check again in the copy-edited version of the paper.

Figure 6: Not necessary but I'm curious if you did both X and Y axes on +- log could the color scale then be linear? Would this presentation show anything different? Useful? Matplotlib has a "symlog" function for this type of display.

We will investigate your suggestion. We are not familiar with symlog.

We did not find that using symlog would help the interpretation of the our figures. We did not change them.

Have you considered and propagated projection errors from your EASE2 grid?

We are not sure what you mean. The EASE2 grid and projection is fully defined and there are no propagated projection errors into the product. Both the satellite TB images and the sea-ice drift vectors are on the same EASE2 projection, so there are no uncertainties there. Please explain what you meant in more detail if our answer is not sufficient.

Regarding my comment about propagating projection errors - this may not be an issue if all your inputs and outputs are on the same grid. I was referring to the fact that, for example, in Greenland EPSG:3413 has 8 % errors with respect to the real world, so if you take 1 m in EPSG:3413 and use that value on another projection, it may no longer be 1 m.

We understand. All our processing steps use the same EASE2 projection so we do not think projection errors occur. No change to the manuscript.

I don't think a recommended citation format is appropriate. Please remove. That's not your decision, that's the decision of the place where the citation is used.

We think this can be useful for a potential user. We agree that the publisher or citation managing tool will define a citation format (hence our suggestion to use doi2bib to get a BibTeX entry). But there are several other situations where one could want to cite the product, e.g. in a presentation, a project report, a grant proposal,.... without a strict citation format. If you agree we will write something like: "In case no citation format is prescribed, we invite you to cite the dataset as: ". But we can also remove it if you are not convinced.

We removed the first recommended citation but kept that in section 8 "Code and data availability" where it is now introduced with the sentence: "In case no citation format is prescribed, we invite you to cite the dataset as:".

Beyond these issues I raise, please carefully consider comments from reviewers. In particular, I too am curious about the motion vs. displacement issue raised by R2. If you prefer displacement over motion for some reason, would it be possible to provide a 'derived' motion product? I am concerned about the complications and user-error issues raised by R2.

This comment by R2 was very relevant indeed and we will give it our full attention when revising the manuscript (see our answer to R2 for details). We will also publish a notebook so that users can directly generate the 'derived' product with u/v velocity convention.

We believe our revised manuscript (with the new section 4.2 and the notebook) addresses R2's legitimate concern.

I now better understand your uncertainty and flags based on your response to R2 (top of p. 9 of your response; also p. 11). From this I wonder about your uncert_dX_and_dY value. Is this variable is space and time, considering higher uncertainties when extrapolation occurs, vs. interpolation, vs. dense observations?

Yes, our uncert_dX_and_dY are 2D fields varying in space and time. To make sure this is well understood, we added some sentences at the end of Section 4.1:

Variable {\tt uncert\_dX\_and\_dY} holds fields of retrieval uncertainties. *These are 2D fields of uncertainties that vary on a daily basis as a result of the processing steps, both for the single\=/sensor and the merged product (Sect. 3.5)*. A *single* uncertainty field is provided *in each product file* for both the \dX\ and \dY\ components (hence the name). This is because our method to derive the uncertainties from the statistics of the validation exercise does not support providing different values for the two components at this stage.

---

## Author Response (AR2)

L19: I guess here you don't need a hyphen since you are not refering to the noun "sea-ice drift" but to the action "sea ice drifts". But this may be an issue solved during the editoral process anyways.

Indeed. We will remove the hyphen here.

L23: "southern" --> "Southern"

Done.

L83: "We use these ..." --> Perhaps better: "We use brightness temperature measurements at these ..."?

Yes. Done.

L124/125: I am a bit confused about the data sources mentioned for the Southern Hemisphere. Perhaps this confusion is driven by Figure 1 where I find buoy trajectories also plotted for the Southern Hemisphere for the time period covered by the Atlas mentioned. My understanding of an Atlas would be that it contains maps of sea ice motion. I am hence wondering whether and how you read and then translated the buoy position information from the Atlas to produce Figure 1. Could it be that these are basically IBAP buoy trajectories - at least for those years that are not covered by AWI buoys?

This is a good question. The Atlas of Antarctic Sea Ice Drift, developed by Schmitt and colleagues at the Karlsruhe Institute of Technology (KIT), and now archived at the Alfred Wegener Institute (AWI) contained both satellite, wind-, and buoy-derived sea-ice motion fields. We only use the buoy trajectories that were made available as part of the Atlas (https://data.meereisportal.de/eisatlas/HTML/eisatlas_buoys.html). The buoy trajectories are indeed from IPAB and other investigators (https://data.meereisportal.de/eisatlas/HTML/eisatlas_references.html).

We now changed our sentence to read (added text in italics): For Antarctic sea\=/ice we rely on two main data sources: the *buoy trajectories compiled as part of the* Atlas of Antarctic Sea Ice Motion \citep{www:karlsruhe-antarctic-icedrift} (1979\=/2000) and the…

Figure 1: Please check whether you need a title above every map. My impression is that this title is not really needed.

The title above each panel is indeed not strictly necessary, we removed them.

L210: "In in" --> "In"

Done.

L225: "force" --> "forces"

Done

L227: "successfull" --> "successful"

Done

Indeed. Done.

Done

Done

Done

L519: Is there a reason why you chose the directions to be opposite to the meteorological notation of u and v components of the wind vector?

There were several issues with our sentence, thank you for pointing this out. We change to:

e.g. latitudinal (aka south-north or meridional) and longitudinal (aka west-east or zonal) components.

Done

Done

Done

Done

We could, but we will keep it as it also allows the reader to quickly locate the table he/she is looking for. I also allows to keep the same structure for all tables.

- My reading of ">= -25km" would be all values larger than -25km, i.e. -24km, -23km, -22km, and so forth. Would it make sense to use ">= |+/- 25km|"?

We changed to read "(absolute value larger than 25 km)" to avoid any confusion.

- "1:1 line" --> So far you used "1-to-1 line"

We changed to "1-to-1" line.

- Why do we have difficulties to resolve highly dynamic drift conditions with comparably fine spatial resolution? Wouldn't this be a function of the search window used?

We discuss this further in section 6.1. There are two main aspects to take into account for this underestimation of the highly dynamic drift conditions: 1) the very rapid drift (as measured from buoys) is not representative of the area averaged drift, and 2) our search window is too small for capturing them. The first one relates to imaging spatial resolution (we cannot have much smaller sub-windows with this type of imaging resolution). The second relates to the search window. In section 6.1 we discuss both aspects without being able to rule any of them out. Still, at this stage of the manuscript, we think it is reasonable to mention imaging resolution as one of the explanations for the underestimation. We thus did not change the manuscript.

Table 7 and 8, values of N: Just for confirmation: The fact that the values of N for season Y is not the sum of the values of N for seasons W and S can be explained by the two missing months of the shoulder seasons spring and fall, right?

Indeed.

L680: I suggest to connect once again the Tschudi et al (2020) data set with the NSIDC data record. I can be guessed that you refer to the same product but it is not entirely clear.

We changed to: "Tschudi et al. (2020) present *the NSIDC sea-ice motion dataset v4 that starts in 1978*".

L699/700: "since all these ... 12.5km" --> In my eyes this is a quite general statement. Especially the NSIDC data product also makes use of the 37 GHz data - especially for SMMR and for the period when SSM/I near 90GHz data were not available - but also in general the Tschudi et al paper states that the drift is derived from both near-90 GHz and 37 GHz data for SSM/I and SSMIS. Hence, there input data partly even have a coarser than 12.5 km resolution.

We rephrased to: However, since *passive microwave imagery from SSM/I and SSMIS with a resolution of 12.5~km at best is the core source of all these data records*, …

Another comment you could make is that the OSI-455 data set is MUCH more homogeneous in terms of the statistics of the input data when comparing the data for both hemispheres. The NSIDC product is based on a highly varying suite of input data, including buoy and AVHRR data for varying periods and coverage in the NH while none of these data are used in the SH. The information content included in their SH drift estimate is therefore considerably less than for the NH. And with that the OSI-455 product is much more consistent and of comparable quality (within the limitations of the approach) for NH and SH - a positive characteristics when using the OSI-455 data set for global model comparisons.

We agree and added the following sentence in section 5.1 (Temporal Coverage): *One of the positive characteristics of the OSI~SAF sea\=/ice drift CDR presented here is thus that the same sources of sea-ice motion are used in the Arctic and the Antarctic, and that the type of satellite imagery is stable through the three decades covered.*

L705/706: A good place to state that you include the information about this variable duration along with the motion product so that the user can check.

We agree and changed to: The duration of each drift vector is roughly 24h from \utctime{12} to \utctime{12}, *but the actual start and end times (and thus durations) of each vector is provided in the product file, since* they vary across the product grid *and* from one day to the next (see \fref{fig:example_dt})

L708: At the same time the weekly product does contain not a single estimate about the quality of the product. This information is only accessible to those users that chose to use the daily product.

This is correct, but we do not feel this is the right place to bring this. We did not change the text.

L762: You could add that this bridging the summer season gap is something also done in the NSIDC data product, combining wind-driven ice motion and buoy motion.

We could, but this was already stated several times, e.g. in section 5 (Comparison to existing CDRs) and in Table 9. We did not change the text.

L767: "bu" --> "but"

Done.

L774: "an" can be deleted?

Indeed. Done.

L791 in context of L781-784: Please check once again whether you really want to make this rather strong statement that your summer sea-ice drift fields ARE generally biased. How significant are the changes reported by Brunette et al.? How much might their conclusions be colored by deficiencies of the PIOMAS model to simulate the full sea-ice thickness distribution correctly (large sea ice thickness is under-, low thickness over-estimated) and other (hypothetical) trends in their forcing data?

The changes reported by Brunette et al. are significant. They indeed use PIOMAS and ERA5, two reanalysis products, which might present biases and trends. However since they tune their free-drift models against on-ice buoy velocities, we can expect that at least part of the biases and trends in SIT are absorbed by the tuning. Concerning the under-ice currents in the Beaufort Gyre region, their results confirmed estimates observed from independent approaches (e.g. mean ocean topography by altimetry). The ramp-up of sea-ice motion was also documented by e.g. Sumata et al. (2023).

Sumata, H., de Steur, L., Divine, D.V. et al. Regime shift in Arctic Ocean sea ice thickness. Nature 615, 443–449 (2023). https://doi.org/10.1038/s41586-022-05686-x

Being a data producer is a delicate balance. One does not want to over- nor undersell one's dataset. We agree with the reviewer that this wording is possibly too strong compared to what we know and compared to the wording we adopted at the beginning of this very section. We thus propose to revise as: "In conclusion, we bring to the attention of the users that our summer sea-ice drift fields might be biased over the 30-years period. "

L794: "our summer .. less biased ..." --> I am ready to second this statement but was wondering what the impact of the buoy drift that is combined with the wind-drift in the NSIDC product could be in this context.

Our statement about the summer bias of the NSIDC product directly stems from the results of Brunette and co-authors (and Sumata et al., 2014). According to Tschudi et al. (2020) (Supplement, lines 36 to 43), the sea-ice drift fields derived from passive microwave enter the final analysis during summer, with the same weights as they have during winter. This might also be an explanation for the low bias of that product during summer. We dit not change our text.

L802-807: This is interesting. It means that your evaluation statistics excludes buoy observations in the Fram Strait and Greenland Sea. Is this the same for the studies that report on the evaluation of the NSIDC product mentioned further up?

No, it is not the same. The impact of excluding buoy observations in the Fram Strait and Greenland Sea is not large, except in months when a lot of trajectories exit the Arctic Ocean through this gate, e.g. the buoy array from the MOSAIC campaign.

L811: "without onwards" ???

We removed "without" and kept only "onwards".

L846: "This is because" --> suggest to add something like "probably", "likely" or the like because at the moment you cannot be 100% certain that this is indeed the reason for your observation.

We added "possibly".